



# Numerical Assessment of Morphological and Hydraulic Properties of Moss, Lichen and Peat from a Permafrost Peatland

Simon Cazaurang[1], Manuel Marcoux[1], Oleg S. Pokrovsky[2,3], Sergey V. Loiko[3], Artem G. Lim[3], Stéphane Audry[2], Liudmila S. Shirokova[2,4], Laurent Orgogozo[2]

[1] Toulouse Institute of Fluid Mechanics (IMFT), National Polytechnic Institute of Toulouse, Toulouse, F-31400, France.
[2] Geosciences Environnement Toulouse (GET) Laboratory, University Toulouse III – Paul Sabatier, Toulouse, F-31400, France.
[3] BIO-GEO-CLIM Laboratory, Tomsk State University, Tomsk, Russian Federation.
[4] N. Laverov Federal Center for Integrated Arctic Research of the Ural Branch – Russian Academy of Science, Russian Federation.

*Correspondence to*:

Simon Cazaurang
Institut de Mécanique des Fluides de Toulouse
2, Allée du Professeur Camille Soula
F-31400 Toulouse, France
mail: simon.cazaurang@imft.fr
tel: +33 534-322-956

**Abstract.** The hydraulic properties of ground vegetation cover are important for high resolution hydrological modeling of permafrost regions, due to its insulating and draining role. In this study, the morphological and effective hydraulic properties of Western Siberian Lowland ground vegetation samples (lichens, *Sphagnum* mosses, peat) are numerically assessed based on tomography scans. After numerical pre-processing, porosity is estimated through a void voxels counting algorithm, showing the existence of representative elementary volumes (REV) of porosity for most samples. Then, two methods are used to estimate hydraulic conductivity depending on the sample's homogeneity. For the most homogeneous samples, Direct Numerical Simulations (DNS) of a single-phase flow are performed, leading to a definition of hydraulic conductivity related to REV, which is larger than those obtained for porosity. For more heterogeneous samples, no adequate REV may be defined. To bypass this issue, a pore network representation of the whole sample is created from computerized scans. Morphological and hydraulic properties are then estimated through this simplified representation. Both methods converged on similar results for porosity. Some discrepancies are observed in the morphological properties (specific surface area). Hydraulic conductivity fluctuates by two orders of magnitude, depending on the method used, and yet this uncertainty is less than that found in experimental studies. Therefore, biological and sampling artifacts are predominant over numerical biases. Porosity values are in line with previous values found in the literature, showing that arctic cryptogamic cover can be considered as an open and well-connected porous medium (over 99% of overall porosity is open porosity). Meanwhile, digitally estimated hydraulic conductivity is higher compared to previously obtained results based on field and laboratory experiments. This could be related to compressibility effects, occurring during field or laboratory measurements. Thus, some



supplementary studies are compulsory for assessing syn-sampling and syn-measurement perturbations in experimentally estimated, effective hydraulic properties of such a biological porous medium.

[Table 1 location]

## 1. Introduction

Covering a quarter of the Northern Hemisphere's land surface (Brown et al., 1997), permafrost soils are the most representative soil types in arctic and subarctic regions. Permafrost is a soil layer in which temperature remains below zero degrees for at least two consecutive years, thus holding ice in its porous structure. Frozen layers make permafrost hydrology peculiar, resulting in complex couplings between heat and water fluxes (Grenier et al., 2018; Tananev et al., 2020). Seasonal structural variations occur in permafrost soils, as surface thawing forms an 'active layer'. Most permafrost-related

biogeochemical processes (especially organic matter degradation) take place in this layer. The active layer is at its maximum thickness in the early autumn, and is generally meters in scale (Clayton et al., 2021; Aalto et al., 2018; Guo & Wang, 2017). Active layer thickness is, nonetheless, spatially variable due to climatic conditions, land cover, and the micro and macro-topography. The impact of hydrological climate change is particularly drastic in permafrost-dominated environments because of deepening thaw fronts (Hinzman & Kane, 1992). Indeed, between 2008 and 2016, the average annual temperature

of arctic permafrost soil increased by 0.4 (± 0.25)°C (Biskaborn et al., 2019; Fox-Kemper et al., in press). This causes positive feedback on average atmospheric temperatures (Meredith et al., 2019), reduces latent heat effects (Walwoord & Kurylyk, 2016) and increase water drainage in Arctic watersheds (Liljedahl et al., 2016). Hence, quantifying heat and water transfer properties in permafrost affected regions is compulsory.

Previous studies have addressed this quantification through field observations (Olefeldt & Roulet, 2014; Streletskiy et al.,

2015; Throckmorton et al., 2016; O'Connor et al., 2020, among others) or field and laboratory experiments (Vedie et al., 2011; Roux et al., 2017; Wagner et al., 2018, among others). Some recent studies have also dealt with this question using a modeling approach (Bense et al., 2012; Genxu et al., 2017; Burke et al., 2020; Du et al., 2020; Fabre et al., 2017, among others).

Bryophytes (mosses) and lichens are widely present in tundra and taiga environments. Dominant ground cover consists of *Sphagnum* mosses and lichens in permafrost peatlands (Soudzilovskaia et al., 2013; Volkova et al., 2018). *Sphagnum* mosses are part of the *Bryophyta* plant division, which represents non-vascular plants (without xylem or phloem). *Sphagnum* colonies grow indeterminately from their apical structure, named the *capitula*. Their water content mainly relies on capillary forces maintained by each individual's density (Hayward & Clymo, 1982; Howie & Hebda, 2018). Lichens are not

'vegetation' but consist of a symbiotic association between heterotrophic *Fungi* and autotrophic *Algae.* Both *Sphagnum* and lichens can be gathered into the *Cryptogamae* subkingdom. This cryptogamic layer has an important impact on permafrost



dynamics because it influences heat and water exchanges between the soil and atmosphere (Soudzilovskaia et al., 2013; Launiainen et al., 2015; Porada et al., 2016; Park et al., 2018; Loranty et al., 2018). Boreal vegetation is assumed to be a major nutrient and inorganic solute exchange medium at a watershed scale (Shirokova et al., 2021). Boreal vegetation is
likely to accumulate in lowlands at a low degradation rate, resulting in the formation of *Sphagnum* peatlands, such as the Western Siberian Lowlands.

Ground vegetation transfer properties are key information for high resolution hydrological modeling of permafrost-related catchments. Thus, reliable estimates of them are necessary for water flux studies for boreal soils and for climate change
impact assessment on the hydrology of high latitude continental surfaces. Therefore, some recent efforts have been put to emphasize the role of the cryptogamic layer in Earth System Models (Stepanenko et al., 2020; Shi et al., 2021). Devoted modeling tools have also been created to predict *Sphagnum* dynamics (*Peatland Moss Simulator* by Gong et al., 2020). Furthermore, specific modeling work has been conducted on restored *Sphagnum* peatlands, to link hydrological properties with dissolved organic carbon dynamics (Bernard-Jannin et al., 2018) or soil moisture dynamics (Elliott & Price, 2020).
However, the mechanistic modeling of water and heat fluxes in ground vegetation layers remains difficult, as their porous media transfer properties are not straightforward to evaluate (Orgogozo et al., 2019).

Many studies are available for the decayed *Sphagnum* layer: peat. The hydrological and thermal properties of peat are well documented. Extensive reviews of the relation between hydrogeochemical processes in peatlands and peat's porous medium
structure were conducted by McCarter et al. (2020). A study of peatland's hydraulic properties was initiated during the 1920s, for peatland drainage (Malmström, 1925). Then, some introductory field experiments were conducted on Finnish peatlands (Virta, 1962; Heikurainen, 1963; Sarasto, 1963) as well as in the United States (Boelter, 1968) and Ireland (Galvin, 1976).

Only a few studies were conducted on the living part of this upper permafrost layer. Hence, quantitative assessments of some key hydrological properties of ground vegetation layers are needed, such as total, open and enclosed porosity, hydraulic conductivity and specific surface area. In terms of hydraulic properties, hydraulic conductivity has been assessed in the laboratory using constant or falling-head permeameters (Quinton et al., 2000; Price et al., 2008; Hamamoto et al., 2015; Weber et al., 2017) or via field measurements (Päivänen, 1973; Crockett et al., 2016; this study). The results are presented in
Table 2, with some peat results for comparison. Otherwise, arctic lichens have received little attention, to date. To our knowledge, only one study has estimated lichens' hydraulic properties, considering unsaturated hydraulic conductivity without taking into account macropores (Voortman et al., 2014). However, the specific surface areas of some other lichen species are documented in the literature (Adamo et al., 2007). Some studies quantified arctic lichen properties in response to acid rain (Tarhanen et al., 1999), to clarify their interaction with the rhizosphere (Banfield et al., 1999), or in relation to their
albedo properties (Bernier et al., 2011).



[Table 2 location]

This study focuses on evaluating the hydrological transfer properties of representative vegetation types of the Western Siberian Lowlands. To this end, natural samples collected from the Western Siberian Lowlands are digitally analyzed to characterize some morphological and hydraulic transfer properties. Thus, contrary to previous works compiled in Table 2, this study aims to assess the hydraulic properties of lichens and *Sphagnum* mosses by numerical methods rather than experimental measurements. The arctic cryptogamic layer is assumed, hereafter, to represent a complex patch work of biological porous media (Price et al., 2008, Voortman et al., 2014, Hamamoto et al., 2015).

To validate this hypothesis, a thorough analysis of sample homogeneity is carried out, based on porosity, as it is the main driver of flow dynamics in porous media (Koponen et al., 1997, Koponen et al., 1996). This enables the classification of samples according to their homogeneity. Indeed, for homogeneous samples, a smaller sample region can be considered as an effective medium sharing the same properties as the whole sample. Multidimensional porosity description leads to a statistical study of the existence of a Representative Elementary Volume (REV). Two standard porous media modeling methodologies are used throughout this study: Direct Numerical Simulations on computed Representative Elementary Volumes (DNS-REV) and Pore Network Modeling with built-in solver (PNM). The impossibility to collect a substantial number of samples is compensated by a statistical quantification of a REV for each sample. This implies that the REV is smaller than the sample, hence sampling size is chosen to match sizes that were used in previous literature, such as Weber et al. (2017).

## 2. Material & Methods

### 2.1 Sample collection and digital reconstruction

Samples are collected at Khanymey Research Station in Western Siberia (Autonomous District of Yamal-Nenets, Russian Federation) in July 2018. Eight moss samples (*Sphagnum sp.*) are collected, either on moss mounds or in thermokarstic hollows. Additionally, two lichen samples (*Cladonia sp.*, named Lichen1.3 and Lichen2.1) and two peat samples (named Peat2.2 and Peat2.3) are collected. Of the eight *Sphagnum* samples, three are *S. angustifolium* (C.E.O. Jensen ex Russow, named Mound2.4, Mound2.5 and Mound2.6), two are *S. lindbergii* Schimp (named Hollow1.2 and Hollow1.4) and two are *S. majus* (Russow) (C.E.O. Jensen, named Hollow2.7 and Hollow2.8). The last moss sample is *S. lenense* H. Lindb. ex L.I. Savicz (named Mound1.1).

Sampling is thoroughly conducted, to minimize structural perturbations. In order to achieve this, each sample's surroundings is cleared with special care prior to extraction. Then, the sample is extracted using a ceramic knife, directly at the right dimensions to fit in a high-density polyethylene box, where it remains from the moment of sampling and drying to the tomographic examination. Additionally, four in-situ hydraulic conductivity measurements are performed on various



*Sphagnum* plots, using a double-ring infiltrometer (Table 2). An overview of the sample collection method is shown in Fig. 1, as well as 3D tomographical visualizations of each sample type.


[Figure 1 location]

The samples are then dried at 40 °C for 48 hours after sampling. Thereafter, each sample is scanned and digitally reconstructed using high resolution X-ray imagery.

X-ray Computed Tomography (X-CT) has been widely studied and is extensively used for medical purposes and geoscientific applications (Christe et al., 2011). Tomography is a non-destructive technique which enables the observation of pore structure data at micron scale, especially for pore space assessment in sedimentary rocks. X-CT scanning has been acknowledged as being an efficient method for accessing morphological information, such as the pore structure of peat soils (Turberg et al., 2014). Cnudde and Boone (2013) published an exhaustive review of X-ray tomography applications for earth

sciences. Rezanezhad et al. (2016) demonstrated that X-CT peat scanning showed a satisfactory spatial resolution for the study of peat's pore morphology. Since bryophytes can be assumed to represent a cluster of individuals, X-CT permits the segmenting of each plant structure, which cannot otherwise be achieved without destructive techniques. Tomographical scans of studied samples are produced using an EasyTom® XL (RX Solutions, France) with a maximal X-ray emission source set to 90 kV. The obtained resolutions, after tridimensional reconstruction, is 94 µm·voxel$^{-1}$, except for 'Lichen2.1' at

88 µm·voxel$^{-1}$ (due to the scanning settings used specifically for this sample). The voxel number ranges from $4.8·10^8$ voxels to $9.6·10^8$ voxels. Virtual samples are cropped and reduced to form a Usable Volume (UV) to avoid sampling border effects. Using ImageJ – Fiji (Schindelin et al., 2012), computed samples are then binarized using an intra-class variance reducing algorithm (Otsu, 1979). This resulted in a sample consisting of an eight-bit black and white image stack. Supplement A contains some of the technical data, such as Usable Volumes and digital reconstructions, for each sample.

**2.2 Drying impacts assessment on sample representativity**

The sampling locations and processing facilities were far away from each other. To ensure structural preservation, special care is taken throughout the sampling, transportation and scanning operations. The samples are oven-dried for 48 hours at 40 °C, at atmospheric pressure, to halt biological degradation. As expected, *Sphagnum* mosses began to whiten and become papery, as described by Hayward & Clymo (1982). The samples are scanned at the IMFT in dry conditions, two months after

their primary extraction at Khanymey Research Station and drying.

To ensure the dry samples' representativity, we reproduced the drying experiment carried out by Kämäräinen et al. (2018). This experiment is conducted on similar *Sphagnum* species (*S. fuscum* (Schimp.) H.Klinggr*., S. majus* (Russow) C.E.O. Jensen) and sampled according to the same method at Clarens mire (Southwest France, N 43°08'41.3", E 0°25'12.9") in

March 2021, notwithstanding that *Sphagnum* acrotelm (growing section of a *Sphagnum* individual) is much thinner than



Siberian samples. Four samples are collected, two being dried at 40 °C for 48 hours and the other two being left untouched, as control samples. Each of the four samples are scanned two days after their extraction and then again, 14 days after extraction. Additionally, one *Sphagnum majus* individual is extracted and left to dry in ambient conditions.

A comparative study between each of the two sample lots and the lone individual show that drying does not affect structural preservation. Our validation experiment converges with the results found by Kämäräinen et al. (2018). This also confirms hyaline cells' structural durability; the early work of Puustjärvi (1977) showed that hyaline cells were well-preserved during biological decay. Drying impacts aside, *Sphagnum's* continuous growth on non-dried control samples seems the most impactful structural factor, as each individual was striving to adapt itself to the sampling box' hydric conditions. Fast drying
before tomographic examination can be a reasonable solution for preserving the morphological structure, in conjunction with careful on-site sampling.

## 2.3 Morphological analysis: Total porosity ($\varepsilon_{total}$), open porosity ($\varepsilon_{open}$), Specific Surface Area $S_{SA}$ and Pore size distribution

Global porosity ($\varepsilon_{Total}$) is calculated for each sample using built-in ImageJ-Fiji's tools and macro scripting. Porosity is
considered to be a ratio between the number of voxels representing the void phase $i_{voxel}$ (void phase's internal volume, including closed porosity) over the total number of voxels representing a sample $N_{total}$ (void and matrix volume). This relation is shown in Eq. (1):

$$\epsilon_{Total} = \frac{V_{i=0}}{V_{Total}} \times 100 = \frac{\int\int\int_0^{N_{i=0}} i_{voxel}(x,y,z)\,dxdydz}{\int\int\int_0^{N_{Total}} i_{voxel}(x,y,z)\,dxdydz} \times 100 = \frac{\sum_0^{N_{i=0}} i_{voxel}}{\sum_0^{N_{Total}} i_{voxel}} \times 100,$$

185                                                                 (1)

Porosity is computed on bi-dimensional horizontal slices along the $z$ axis to evaluate porosity variations along the samples. Image stacks are then reconstructed along the $x$ and $y$ axis, to create two other image stacks. Finally, porosity is computed along the $x$, $y$ and $z$ axis using a voxel counting algorithm, shown in 3.1.

The samples could then be classified into three types, according to the porosity profile along the vertical axis (Table 3 and
Fig. 4). As porosity appears to be almost constant over the $x$ and $y$ axis, sample classification is solely based on vertical porosity $z$:

- Type I: Constant high porosity along $z$ axis excluding border effects;
- Type II: Low basal porosity, linearly increasing to the top of the sample;
- Type III: No specific trend observed on vertical porosity.



Open and connected porosity ($\varepsilon_{open}$) are retrieved using dedicated shape analysis and labeling tools provided in the IPSDK™
image processing toolkit (a Reactiv'IP product, used in Goubet et al., 2021). This enables a precise segmentation to associate
each connected void space into a unique identifier. Here, since the samples have more void than matter, this first label is
assumed to be connected void space, which plays a major role in the flow and transfers (porosity), the latter being a closed or
non-communicating element. From the raw dataset, voxel intensity is integrated to get the first label's voxel sum  divided by
the overall voxel number, as shown in Eq. (2):

$$p_{open} = \frac{\epsilon_{open}}{\epsilon_{Total}} \times 100 = \frac{\sum_{0}^{N_{i \in label 0}} i_{voxel}}{\sum_{0}^{N_{Total}} i_{voxel}} \times \epsilon_{Total}^{-1} \times 100,$$

(2)

The specific surface area is deduced using the same shape analysis and labeling tools included in IPSDK™. Integrating the
surface between both phases (void and solid) yields the total surface $S$. Thus, volumetric specific surface area $S_{SA}$ is obtained
by dividing this surface with the sample's bounding box volume, expressed in $m^2.m^{-3}$, as shown in Eq. (3):

$$S_{SA_V} = \frac{S_{solid}}{V_{BBox}} = \frac{\iint_{0}^{N-1} S\big((i,j)(i+1,j+1)\big) di dj}{L_X L_Y L_Z},$$

210                              (3)

Specific surface area is conventionally expressed in relation to density (in $m^2.g^{-1}$). For this purpose, each dried sample mass
is obtained using an analytical balance and the sample's dry bulk density $\rho_{dry}$. Then, volumetric specific surface values are
converted into a mass-related specific surface by dividing volumetric specific surface area with dry density, as shown in Eq.
(4):

$$S_{SA_M} = \frac{S_{SA_V}}{\rho_{dry}},$$

(4)

Pore size distribution is calculated using ImageJ-Fiji's implemented image segmentation tools on the binarized image stacks.
On each stack's image, a Euclidean distance transformation of the matrix phase from the void phase is first applied. Then, for
each isolated void patch, the Feret diameter is computed.




**2.4 Darcy scale morphological and hydrological properties' definition: Representative Elementary Volume (REV)**

In this study, the collected samples are assumed to form a complex fibrous porous media. Resolving mechanistic equations in such large domains is not straightforward due to the extensive computational resources required. Conversely, resolving such equations on an arbitrary cropped sample would not aid the hydraulic property assessment. To make the link between microscale and macroscale phenomena, a reproducible pattern is required to avoid microscale heterogeneities and lack of information due to a diminutive sample size. To do this, finding a representative region that validates scale separation assumptions with both microscale and macroscale heterogeneities is compulsory, thus defining the volumetric average of a microscale property that is continuous and informative at a macroscale. One of the first volume averaging methods consists of finding a statistical Representative Elementary Volume (REV) for the given studied property.

Indeed, REV is a theoretical concept clarifying the definition of the macroscopic scale (Darcy scale) and the microscopic scale (pore scale) and characterizing a given porous medium. This REV can be assumed as a specific sample volume, in which transfer governing equations (single-phase flow, for example) may be defined, along with the associated effective properties. A proper mathematical definition of a REV is given in Bachmat and Bear (1987), Quintard and Whitaker (1989) and Whitaker (1999), along with a thorough definition of volume averaging methods. A generic profile for a given property $\varphi_\beta$ is shown in Fig. 2.

[Figure 2 location]

The fluctuation profile shows three main domains. Here, the REV is defined as the smallest volume for which statistical fluctuations of a given property in a given space are sufficiently low to consider its average value as an effective property. Finding the Representative Elementary Volumes of some key properties (e.g., porosity and intrinsic permeability) is a routine workflow in porous media sciences. It is often used for fractured oil reservoirs (Durlofsky, 1991) or artificially packed glass bead media (Leroy et al., 2008). A REV is, by definition, large if compared to characteristic lengths of heterogeneities at a microscopic scale but small if compared to characteristic lengths of heterogeneities at the macroscopic scale. Thus, the properties computed for a REV of a porous medium may be defined and computed as continuous functions of space and even constant, in the case of a homogeneous porous medium, as defined by Bear (1972). In general, REVs are described on the basis of morphological characteristics such as porosity, although a distinct REV can be found for any given porous media property. Porosity and hydraulic conductivity related REVs are characterized throughout this study, leading to two different sizes, one for each property.

**2.4.1 Porosity: Binarization and voxel counting**

From previously binarized image stacks, a statistical REV analysis is conducted using dedicated high performance image processing Python libraries (IPSDK™), encapsulated in a specifically designed batch process. First, porosity (Eq. 1) is





computed for a given sub-sampling volume within the whole sample. Then, the sub-sampling volume location is incrementally reduced and moved in every spatial direction. For each sub-volume, intermediate porosities are computed. The average and standard deviation are stored for each chosen sub-sampling volume. Then, an algorithmic routine is used to find

the maximal size that satisfies a given threshold (1, 3 or 5% of porosity fluctuation). These thresholds define the statistical representativity of these REVs. Thus, a REV satisfying a one percent threshold can be assumed to be a high-grade REV, whereas the five-percent threshold corresponds to lower-grade REVs. For 10 of the 12 studied samples, a REV of porosity is found. The two remaining samples, Hollow1.2 and Peat2.2, do not exhibit a REV for the chosen thresholds.

### 2.4.2 Hydraulic conductivity: Direct Numerical Simulation

Hydraulic conductivity is estimated through single-phase flow computations performed by solving Navier-Stokes equation in the pore space of the considered sample. The concept is to carry out the numerical simulation of fluid flow, reproducing the conditions occurring in a constant-head permeameter. Then, a sample's hydraulic conductivity is computed from the obtained velocity field. A virtual constant-head permeameter is created by imposing a constant pressure on two opposite faces to one direction (inlet and outlet). Watertight wall boundary conditions are applied on other faces, as shown in a

conceptual representation of the initial and boundary conditions (Fig. 3).

[Figure 3 location]

Due to computation time limitations, the biggest studied sub-volume with this approach corresponds to a quarter of the total

sample. In section 2.4, we stated that the REVs of effective physical properties were valid for that particular physical property. Thus, a hydraulic conductivity REV is required, to statistically assess hydraulic conductivity. For that purpose, instead of counting voxel value algorithms (as made for porosity), retrieving a Representative Elementary Volume for hydraulic conductivity requires extensive fluid mechanics simulations. Here, a laminar single-phase flow induced by a pressure gradient is computed for each sub-sample, being consistent with the idea of reducing and moving a defined sub-

volume inside the overall sample. As these simulations are resource-costly, Type I samples (constant porosity) are selected as they are sufficiently homogeneous for the establishment of REVs. Other types are treated by another method presented in 2.4.3. The implemented method relies on Mohammadmoradi & Kantzas (2016), in conjunction with automatic mesh manipulation tools (*trimesh* Python library, Dawson-Haggerty et al., 2019). For each Type I sample, single-phase flow simulation through a fraction of the solid volume (representing a sample) is conducted. The computation is based on the

SIMPLE algorithm (*Semi-Implicit Method for Pressure-Linked Equations* - Patankar, 1980) nested in the *simpleFoam* solver of the open-source Computational Fluid Dynamics toolkit OpenFOAM (Weller et al., 1998, www.openfoam.org; www.openfoam.com).

For each sample, four potential REV sizes are computed (23.5 mm, 15.7 mm, 11.8 mm and 9.4 mm), consisting of 8, 27, 64 and 125 simulations on the *x*, *y* and *z* axis, respectively, representing 672 simulations per sample. This is run on the tier-2



supercomputer *Olympe* (CALMIP computational mesocenter, Toulouse, France). These calculations are run simultaneously, each occupying one node (36 physical cores), representing 10,500 h CPU (about 12 days of physical time) per sample. For each simulation, the velocity field $u_i$ is integrated with the overall outlet surface $S_{outlet}$ (including the surface occupied by the solid matrix) to get an averaged outlet flux value $v_i$, according to Eq. (5):

$$v_i = \frac{1}{S_{outlet}} \int_0^S u_i \, dS,$$

(5)

A careful convergence study is also conducted so that numerical errors, associated with discretization resolutions and iterative procedures for the approximated inversions of the linear systems involved are low enough to be neglected in the analysis of the results. Inlet pressures are chosen to avoid turbulent flows (Re << 1). The computed Darcy velocity $v_i$ could

then be injected into a regular Darcy's law, as shown in Eq. (6), where $k_{ii}$ is a tensorial component of intrinsic permeability (m²) and $\mu_w$ is the dynamic viscosity:

$$k_{ii} = v_i \frac{\mu_w}{\nabla P} \text{ with } \nabla P = \frac{P_{inlet} - P_{outlet}}{L_i},$$

(6)

To avoid artifacts related to the physics of a specific fluid, the *simpleFoam* solver uses kinematic pressure (expressed in $m^2.s^{-2}$) and kinematic viscosity $\nu$ (in $m^2.s^{-1}$) to solve Navier-Stokes equations. These equations are based on intrinsic permeability $k$ expressed in m². However, in the field of hydrology the hydraulic conductivity in m.s⁻¹, abbreviated to $K_w$, is generally used. One can relate hydraulic conductivity $K_w$ to intrinsic permeability $k$ by using Eq. (7) described by Claisse (2016).

$$k = \frac{K_w \mu_w}{\rho_w g},$$

(7)

In continental surface hydrology, liquid water's physical property variations (e.g., volumetric mass $\rho_w$ and dynamic viscosity $\mu_w$) are generally neglected. Thus, intrinsic permeability values obtained from the numerical computations were converted

using water's thermodynamic properties at 293.15 K and 1.013 kPa (Chemical Rubber Company & Lide, 2004), considering the following conversion equation (Eq. (8)):

$$k = 1.0217 \cdot 10^{-7} K_w,$$

(8)





This method is suitable for samples meeting porosity homogeneity requirements, classified into Type I samples. However, another method is needed to compensate for Type II and Type III samples' heterogeneity, as using Direct Numerical Simulations on a complete Usable Volume is prohibitive, in terms of computational resources.

### 2.4.3 No REV for hydraulic conductivity: use of Pore Network Modeling (PNM)

For the samples that do not exhibit a REV for hydraulic conductivity (Type II and Type III samples), the hydraulic
conductivity is then studied using a Pore Network model, generated from the binarized image stacks. Pore Network models are based on the structural simplification of a complex pore structure (rocks or reactive porous industrial media, for example) into a two-state model: spheres and throats. This method often uses various image processing and segmentation tools to generate a network of spheres and linking throats, based on an initial tridimensional volume. Introduced by Fatt (1956), pore network modeling was first studied in conjunction with predefined network properties. Then, pore network generation was
adapted to model some porous media, scanned with X-ray tomography using image processing algorithms, as accurately as possible (Dong & Blunt, 2009, among others). Various algorithms are used to create the internal pore network structure, such as the *maximal ball* algorithm (Silin & Patzek, 2006). More recently, other algorithms based on the morphological properties of the studied porous media have emerged, such as the *Sub-Network of the Oversegmented Watershed (SNOW)* algorithm (Gostick, 2017). This alternative algorithm is considered to be computationally efficient, allowing a porous medium to be
accurately modeled by numerical imagery (Khan et al., 2020). The SNOW algorithm showed a good fit with the standard *maximal ball* algorithm. Generating a pore network and simulating a flow in it is often cheaper, in terms of computational resources, when compared to Direct Numerical Simulation. However, more complex transfer mechanisms, such as imbibition and drainage, are still in the study phase and some extensive work on computational optimization has yet to be conducted, specifically on non-user-generated porous media (Maalal et al., 2021).


For each binarized type II and type III image stack, a direct pore network extraction is conducted using the SNOW algorithm, implemented in the *OpenPNM* and *PoreSpy* open-source Python libraries (Gostick et al., 2019). Then, a synthetic porosity and a synthetic specific surface area may be computed for the obtained simplified representation of the pore space of the porous medium. Using the implemented Stokes' equation solver, a diagonal permeability tensor is retrieved from the
generated pore networks, applying the identical boundary conditions, as in Fig. 2, based on the method given by Sadeghi & Gostick (2020). Once again, intrinsic permeability tensors are converted into a hydraulic conductivity tensor using the relation in Eq. (8).

In Supplement C, a comparative study is described, based on Type I samples between both developed workflows. Then,
some clues are given as to whether Direct Numerical Simulation (DNS) or Pore Network Modeling (PNM) is suitable for a given sample. This comparison shows that Pore Network Modeling is suitable to bypass the heterogeneity issues observed in our samples. Indeed, the obtained porosity values with PNM are in a five-percent threshold, compared to voxel counting





results (Eq. (1)). The hydraulic conductivities computed by PNM and DNS are more contrasted, with one to two orders of magnitude of difference. One should bear in mind that the range of hydraulic conductivity of natural porous media is huge, with up to fifteen orders of magnitude between coarse gravel ($10^{-1}$ m.s$^{-1}$) and unweathered shale ($10^{-15}$ m.s$^{-1}$). Besides this, it is logical that the simplifications involved in the PNM method result in information loss compared to the DNS method. On the other hand, computational time savings (by using the PNM method) are huge (counted in tenth of days for DNS and hours for PNM). In some cases, (e.g., samples of Type II and III) DNS is simply not possible with the current regional scale supercomputing means.

## 3. Results

### 3.1 Morphological analysis

The global porosity and open porosity proportion ($p_{open}$) for each sample is shown in Table 3, ranging from 40% to 50% (samples Peat2.2 and Peat2.3) to more than 95% (for *Sphagnum* sample Hollow2.7).

[Table 3 location]

On average, lichens are the most porous of the collection and peat is the least porous. Porosity values are in line with previously obtained data from the literature, for the highest porous media of the collection (Yi et al., 2009). However, an important variability can be observed for *Sphagnum* samples, gathering minimal and maximal porosity values. Mound mosses have an average porosity of 65.9 ± 22.3%, whereas, average hollow moss sample porosity is 79.6 ± 20.2%. Porosity profiles for each sample are presented in Fig. 4.

[Figure 4 location]

No specific trend can be accessed from the *x* and *y* axes porosity profiles and yet, variations can be observed on the *z* axis. Again, three trends can be observed, clustering samples into three groups according to their respective porosity profile trends:

- Type I: Stable high porosity profile samples, excluding border effects ($\varepsilon_{total} > 85\%$): Mound2.6, Hollow2.7, Hollow2.8, Lichen2.1, Lichen1.3;
- Type II: Medium high porosity profile samples associated with a progressive increase from the bottom to the top ($70\% \leq \varepsilon_{total} \leq 85\%$): Hollow1.2, Hollow1.4, Peat2.2, Peat2.3; and
- Type III: Medium low porosity ($\varepsilon_{total} < 70\%$) associated with no specific trend porosity profiles: Mound1.1, Mound2.4, Mound2.5.

Type I class contains both lichen samples (Lichen2.1, Lichen1.3) whereas type III only consists of mound *Sphagnum*
(Mound1.1, Mound2.4, Mound2.5). Type II class contains half of the hollow *Sphagnum* samples, as well as both peat
samples (Peat2.2, Peat2.3).

Open and connected porosity ($p_{open}$, Table 3) represents nearly all the void space volume in each sample. Open porosity ratio
values range from 0.99 to 0.9999. Thus, we can assume that, due to the fibrous nature of the studied material, enclosed
porosity is not playing a major role in the flow dynamics of the studied samples.

Pore size distribution (Fig. 5) is heterogeneous in each sample and the sizes are concentrated between 0.01 mm and 1,00 mm
of pore radii. The median pore size varies from 0.23 mm (for peat samples) up to 0.88 mm (for lichen samples).

[Figure 5 location]

Intermediate median pore size values can be found for mound *Sphagnum* samples at average values between 0.34 mm and
0.70 mm (for hollow *Sphagnum* samples). According to the previous classification, the median pore size for each sample
type (I, II and III) is 0.66 mm, 0.42 mm and 0.33 mm, respectively. While Type II and Type III curves differ in
bidimensional porosity along *z*, they share similar global pore size distributions. Type I samples are distinct from Type II and
Type III curves. Specific Surface Area ($S_{SA}$) values for each sample are shown in Fig. 6.

[Figure 6 location]

Specific Surface Area values seem to be uneven between each sample type. For instance, low specific surface areas can be
observed for some hollow *Sphagnum* samples ($2.6 \cdot 10^{-2}$ m$^2$.g$^{-1}$ and $2.9 \cdot 10^{-2}$ m$^2$.g$^{-1}$ for Hollow2.7 and Hollow2.8, respectively).
Higher specific area values can be found for one mound of *Sphagnum* ($2.0 \cdot 10^{-2}$ m$^2$.g$^{-1}$ for Mound2.4) and for one hollow
*Sphagnum* sample ($1.7 \cdot 10^{-1}$ m$^2$.g$^{-1}$ for Hollow1.2). In Supplement C, the comparison between results obtained with image
processing and PNM shows that specific surface area seems to be overestimated with PNM, with values mostly higher than
values obtained by face counting (Eq. (3)).

**3.2 Porosity**

Representative Elementary Volumes for porosity have been computed when possible. For samples exhibiting a REV,
porosity has been computed using Eq. 1 applied to the REV. For samples admitting no REV, porosity has still been
computed using Eq. 1, but applied to the whole usable volume of the sample. A REV retrieval algorithm was applied to all
the twelve studied samples, although two of them (Hollow1.2 and Peat2.2) did not admit a REV. Obtained REV sizes are



shown in Table 4. Some examples of tridimensional visualizations of REVs of Porosity are shown in Supplement A and tridimensional porosity plots are available in Supplement B1.

[Table 4 location]


REV$_\varepsilon$ sizes vary from 2 mm to 2 cm, representing $8.0 \cdot 10^3$ to $2.19 \cdot 10^6$ voxels. Substantial morphological variations are visible, spanning from simple tubular structures (visible in the REV of Hollow2.7) to a complex and fibrous medium (for the Mound2.6 sample, Supplement A). The average porosity obtained for these REVs varies from 83.4% to 96.0%, which confirms the high porosity factor of these biological media. Computation times for porosity-REV retrieval ranges from three

to six hours, using two Intel® Xeon® E5-2680 v2 (2.80 GHz) processors and 128 GB of RAM, using high performance Python image processing libraries (IPSDK™).

Graphical synthesis of the digital porosity assessment is presented in Fig.7.

[Figure 7 location]

**3.3 Hydraulic conductivity**

Due to the time and computational resources needed to achieve a careful study of a Representative Elementary Volume of hydraulic conductivity, only Type I samples were studied by DNS, as they represent the most homogeneous samples of the collection. Computed REVs of the hydraulic conductivity sizes are given in Table 5. Diagonal hydraulic conductivity tensor components are shown in Fig. 8 and box plots are available in Supplement B2. Computations for the largest sub-sample size

(on a 23.5 mm edge), showed component hydraulic conductivity values than for the three smaller sizes. This discrepancy can be related to an insufficient computation number for obtaining a good average value, hence a wider statistical spread around the mean value. Moreover, the higher values for the largest studied sizes can also be correlated to heterogeneous hydraulic conductivity behavior, as theoretically shown in Fig. 2, such as effects related to the existence of macropores. An example of an obtained pressure field on a sub-sample of Hollow2.8 through DNS, is shown in Fig. 9-Left.


[Table 5 location]

For three of the Type I samples, REV$_K$ length is computed as 15.7 mm, which is the second largest computed size. Variations in hydraulic conductivity, with respect to study volume reduction, are smaller than those found for porosity, although study

points were scarcer in the case of hydraulic conductivity assessment. Lichen1.3 shows the smallest REV$_K$. It can be seen that the smallest REV$_\varepsilon$ was also described for Lichen1.3. Size differences can be seen between REV$_\varepsilon$ and REV$_K$, up to five times larger for Lichen1.3 and half the REV$_\varepsilon$ for Mound2.6. This seems to show that the Representative Elementary Volume found



for porosity cannot accurately describe properties such as hydraulic conductivity. This is often the case, as porosity REV is smaller than the REVs defined for other properties (Zhang et al., 2000; Costanza-Robinson et al., 2011).


Numerical estimations of hydraulic conductivity are presented in Fig. 8. For each sample of Type I, the axial components of the hydraulic conductivity tensor is given, based on the Representative Elementary Volume of hydraulic conductivity. For Type II and Type III samples, hydraulic conductivity estimates are given, based on pore network modeling. An example of hydraulic conductivity computation is shown on sample Mound2.5 in Fig. 9-Right. Using a pore network allows the

estimation of properties in a model based on the whole sample. The use of a pore network is an affordable alternative to Direct Numerical Simulations at the cost of accuracy.

[Figures 8 & 9 location]

The values obtained, vary from $1.1\cdot10^{-1}$ m.s$^{-1}$ to $9.5\cdot10^{-1}$ m.s$^{-1}$ for Type I samples and from $7.8\cdot10^{-3}$ m.s$^{-1}$ to $4.8\cdot10^{-1}$ m.s$^{-1}$ for Type II and III samples. Type I samples can be assumed to be highly water conductive biological media. Mean hydraulic conductivity decreases when the computed region size becomes smaller, for each direction and each sample (Supplementary section B2). *In-situ* measurements, conducted by infiltration (Table 2), give an average of $10^{-5}$ m.s$^{-1}$, which is in the same order of magnitude as previously published field measurements (Crockett et al., 2016) and computed values (McCarter &

Price, 2012). Analogous values for vertical hydraulic conductivity have been found in the literature at $k_{zz} \approx 10^{-2}$ m.s$^{-1}$ (Päivänen, 1973; Crockett et al., 2016; Golubev et al., 2021). However, other studies showed results of a different order of magnitude for *Sphagnum* samples, with values under $10^{-4}$ m.s$^{-1}$ (Hamamoto et al., 2015). These differences could be explained by the experimental method used to retrieve hydraulic conductivity, as well as *Sphagnum* bog oscillation occurring during sampling (*mire breathing*) (Strack et al., 2009; Golubev & Whittington, 2018; Howie & Hebda, 2018), which is going

to be discussed in the next part.

## 4. Discussion

Digital assessments of the morphological and hydraulic properties of *Sphagnum* and lichens of the Western Siberian Lowlands presented in this work, suggest extremely porous, connected media with high specific surfaces and high hydraulic conductivities. These results are in line with the biogeochemical observations of Shirokova et al. (2021), demonstrating the

overwhelming role of *Sphagnum* mosses in organic carbon, nutrient and inorganic solute fluxes in the Western Siberian Lowlands. Nonetheless, discrepancies between the numerical results presented in this work (Fig.7 and Fig. 8) and previously published measurements of the hydraulic properties of *Sphagnum* are noteworthy (Table 2). Weaker, but still sizeable, differences can be seen between the results given by both of the numerical methods used here for the estimation of hydraulic conductivity on the same sample, namely Direct Numerical Simulation (DNS) and Pore Network Modeling (PNM). This last

methodological point is discussed in Supplement C, where a comparative validation is performed between DNS and PNM on homogeneous samples (Type I).

## 4.1 Numerical reconstruction after scanning

Due to technical limitations, scanning devices have a minimal resolution that causes a loss of information, acting as a threshold. In this study, minimal resolution fluctuated between 88 and 94 µm.voxel$^{-1}$, meaning that two elements of this size

could not be distinguished. In our study, technically unreachable porosity (porosity that is smaller than the minimal scanning resolution) is assumed to play a negligible role in transfers through a saturated medium, reacting as an enclosed porosity. Pre-processing algorithms (especially binarization) can cause information loss due to the arbitrary categorization of each voxel. This erroneous description can be seen for small elements (such as *Sphagnum* leaves) which shrink them. Mesh generation may also bring some additional 'over-erosion' that helps flows inside a sample. These impacts could be studied

by reducing scanning resolution, albeit not available at the time of the scans. However, hydraulic conductivity overestimation in DNS that could be related to these pre-processing effects is likely to be negligible. Indeed, the high porosities encountered and the preferential flow paths that occur in the largest pores (macro-pores) predominate over enclosed pore dynamics. This might not be the case for unsaturated hydraulic property assessments.

## 4.2 Numerical results *vs*. field experiments: porosity and specific surface

As described in previous sections of this study, the samples collected are considerably porous. Porosity values are in line with past results found in the literature (Yi et al., 2009; Kämäräinen et al., 2018), with porosities above 90% for some of the samples. Interestingly, volumetric digital specific surface can be well linked with the porosity of complete samples, as well as the average porosities found for Representative Elementary Volumes.

A clustering can be seen for the three studied sample types (Fig. 10), although mathematical relations between specific surface and porosity are not well-defined for such porous media. The specific surface values obtained are of the same magnitude as previous values obtained for other natural moss and lichen species, using geometrical calculations ($1.4 \cdot 10^{-1}$ m$^2$.g$^{-1}$ for *Hypnum cupressiforme* (Hedw., 1801) moss and $2.4 \cdot 10^{-2}$ m$^2$.g$^{-1}$ for *Pseudevernia furfuracea* ((L.) Zopf, 1903) lichen in Adamo et al., 2007). These values are still notably lower than the values obtained using the B.E.T. method of N$_2$

adsorption isotherms ($1.1 \cdot 10^{1}$ m$^2$.g$^{-1}$ for artificially grown *Sphagnum denticulatum* (Brid., 1926) in Gonzalez et al., 2016). As discussed in Section 4.1, a lack of micropores could explain the observed discrepancies (one to two orders of magnitude) between calculated geometric and B.E.T.

[Figure 10 location]



### 4.3 Numerical results *vs*. field experiments: hydraulic conductivity

The obtained numerical hydraulic conductivities tend to show high, and relatively isotropic, hydraulic conductivity tensor values. Hydraulic conductivities found using Direct Numerical Simulations (DNS) are sizably higher than previous values found in the literature using field percolation (Table 2), often by up to one to three orders of magnitude. The hydraulic conductivities found using Pore Network Modeling seem to be more in line with the values in Table 2. Nevertheless, it should be kept in mind that the results obtained by this method are less structurally accurate that those obtained from DNS, since they rely on a simplified description of the pore structure. Some clues can be advanced to explain this discrepancy: the first being the impact of numerical reconstruction routines and mesh generation procedure (discussed in 4.1); the latter being moss compression during field experiments.

Our digital, constant-head permeameter experiments were conducted in a fully saturated media. Technically unreachable porosity (porosity that is smaller than the minimal scanning resolution) is assumed to play a negligible role in transfers through a saturated medium, reacting as enclosed porosity. In the case of low permeability porous media, such sub-resolution porosity may affect flow (Soulaine et al., 2016). However, in the case of highly porous and connected media like mosses and lichens, the effects related to sub-resolution porosity are assumed to be low, when compared to the effects of the large macropores, which has been shown by Baird (1997). It should also be noted that most of the porosity is opened and connected in our case.

However, moss and lichen samples are compressible (Howie & Hebda, 2018; Price & Whittington, 2010). Field percolation experiments induce a sizeable and rapid mass imbalance on this bryophytic cover, compacting the pore space more than would occur in natural rainfall conditions. This might notably affect flow patterns in macro-pores and explain the lower hydraulic conductivities found in field experiments. Therefore, the numerical hydraulic conductivity assessments carried out in this study enable property quantification of the medium without perturbation, such as compression of the biological pore structure, which is not possible in field experiments.

### 5. Conclusions and perspectives

A numerical assessment of morphological and hydraulic properties was carried out on digital X-CT reconstructions of samples of *Sphagnum* moss, lichen and peat from the Western Siberian Lowlands' bryophytic cover. This porous media-centered approach confirmed the high porosities (from 70 to 95% for most samples) already found in previous studies involving experimental measurements. Hydraulic conductivity estimation was conducted using Direct Numerical Simulations for Type I samples and Pore Network Modeling for Type II and III samples, both fluctuating around $10^{-1}$ m.s$^{-1}$. Indeed, both methods used in this study converge to classify macroscopic lichen, *Sphagnum* moss and peat as being



considerably porous and pervious biological media. Hydraulic conductivity tensor shows isotropic horizontal components, however, some differences can be seen, particularly on the vertical component. Both methods reach higher values than seen before in the literature. This may have been caused by interfering phenomena, such as moss compressibility, occurring during field experiments.


These results provide firm ground for quantitative hydrological modeling of the bryophytic cover in permafrost-dominated peatland catchments, which is crucially important for a better understanding of the global climate change impacts on arctic areas. Using numerical methods potentially enables the assessment of moss and lichen's structural hydraulic conductivity without disturbance by any biological or physical phenomena. Therefore, the porous medium approaches developed

throughout this study lead to unprecedented qualitative and quantitative descriptions of such peculiar, highly porous, biological media.

These physical properties can then be used as input parameters to describe ground vegetation layers in high resolution hydrological models of arctic hydro-systems and extensively refine simulations of this critical compartment of boreal

continental surfaces. For example, they will be used in further modeling studies of permafrost under climate change at the Khanymey INTERACT station, in the framework of the HiPerBorea project (hiperborea.omp.eu). Further studies are needed to assess variable water content consequences on peat and vegetation pore structure. Indeed, water content is one of the main drivers controlling effective transport properties, such as unsaturated flow, volume change and thermal conductivity.

**Acknowledgments**

This work was supported by the French Research Agency (ANR) under the HiPerBorea project (High Performance computing for quantifying climate change in Boreal areas, grant ANR-19-CE46-0003-01), by the CNRS (BryoPhyGel project) and by the French embassy in Russia (PHC Kolmogorov project 2017 N° 38144TB). This work was granted access to the HPC resources of the CALMIP super-computing center, under the allocation 2020-[p12166]. Partial support from the Tomsk State University Development Program («Priority-2030») is also acknowledged. S. Loiko and A. Lim thank the

Russian Science Foundation (project no. 18-77-10045) for supporting the field work. The authors thank L. Bernard and R. Abbal of Reactiv'IP for their support regarding the developments of the digital characterizations with IPSDK™. The authors also want to thank the Clarens mire steering committee (http://tourbiere-clarens.n2000.fr/) for giving us permission to collect samples.



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



**Declarations**

**Funding:** French Research Agency (ANR) - HiPerBorea project (ANR-19 CE46-0003-01) / CNRS (BryoPhyGel project) / French embassy in Russia (PHC Kolmogorov project 2017 N° 38144TB) / Russian Science Foundation (project no. 18-77-10045).

**Conflict of interest:** All authors certify that they have no affiliations with or involvement in any organization or entity with any financial interest or non-financial interest in the subject matter or materials discussed in this manuscript.

**Availability of data and material:** Material is available on the project's website (www.hiperborea.omp.eu).

**Author contributions:**

**Funding acquisition & Project administration:** Laurent Orgogozo.

**Conceptualization, Supervision & Validation:** Manuel Marcoux, Laurent Orgogozo.

**Investigation, Formal analysis, software, Data curation, visualization & Writing original draft:** Simon Cazaurang.

**Resources, Field Methodology:** Serguey Loiko, Artem Lim, Stéphane Audry, Liudmila Shirokova, Oleg Pokrovsky.

**Reviewing:** Manuel Marcoux, Laurent Orgogozo, Serguey Loiko, Artem Lim, Stéphane Audry, Liudmila Shirokova, Oleg Pokrovsky.

 

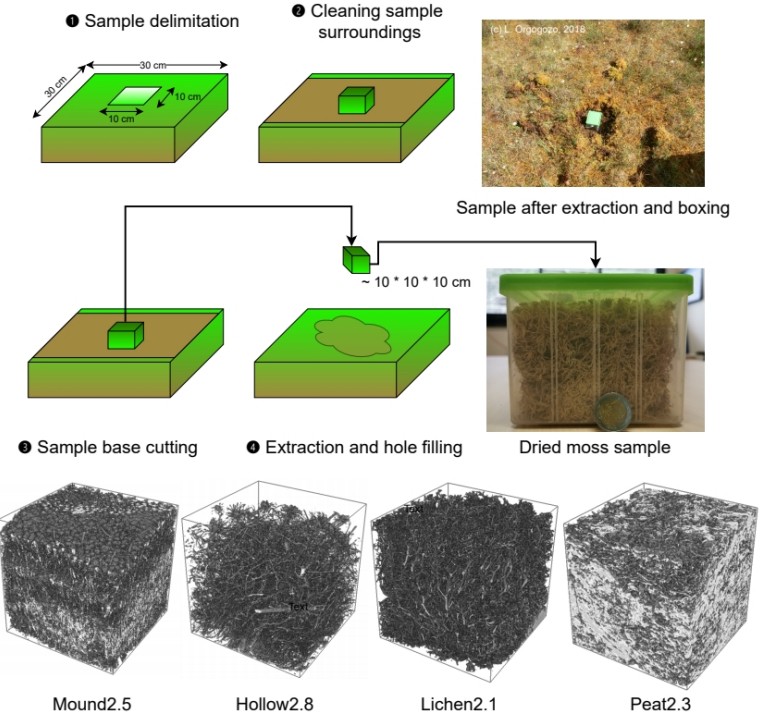

**Figure 1: Sampling collection method overview.**

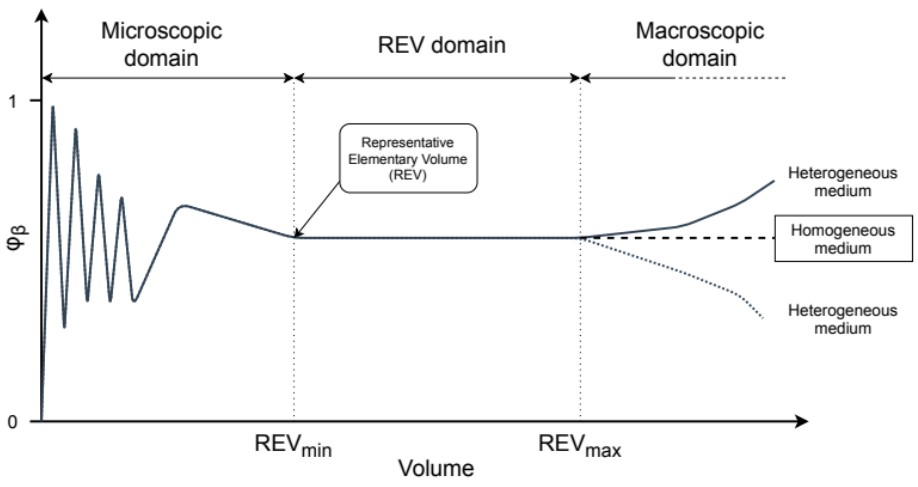

**Figure 2: Schematic representation of fluctuations of a generic property φ_β in conjunction with volume (adapted after Brown &**
945 **Hsieh, 2000).**



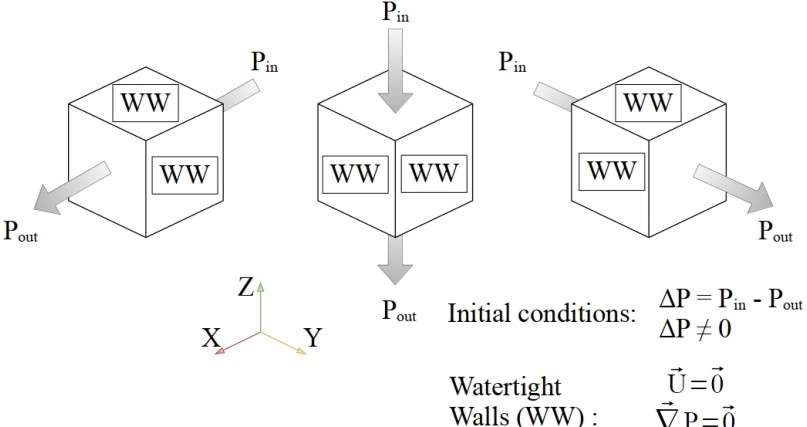

**Figure 3: Initial and boundary conditions used for the Direct Numerical Simulation on sub-volumes of samples and pore network models.**

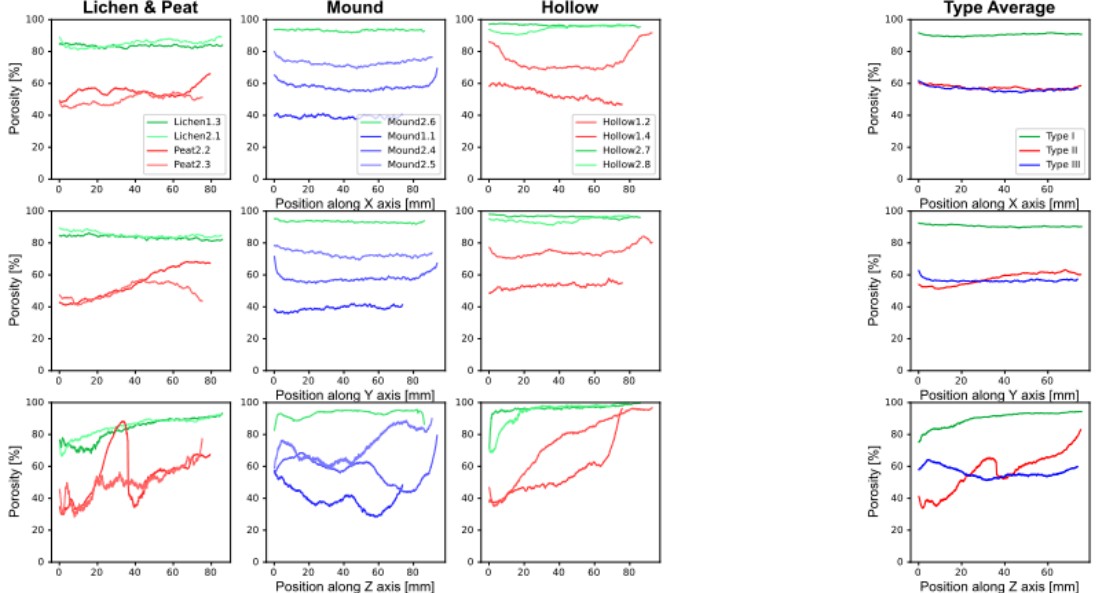

**Figure 4: Planar porosity plot along *x*, *y*, and *z* axis for moss, lichen and peat samples. An averaged value is computed for each sample type, each color nuance representing each type.**



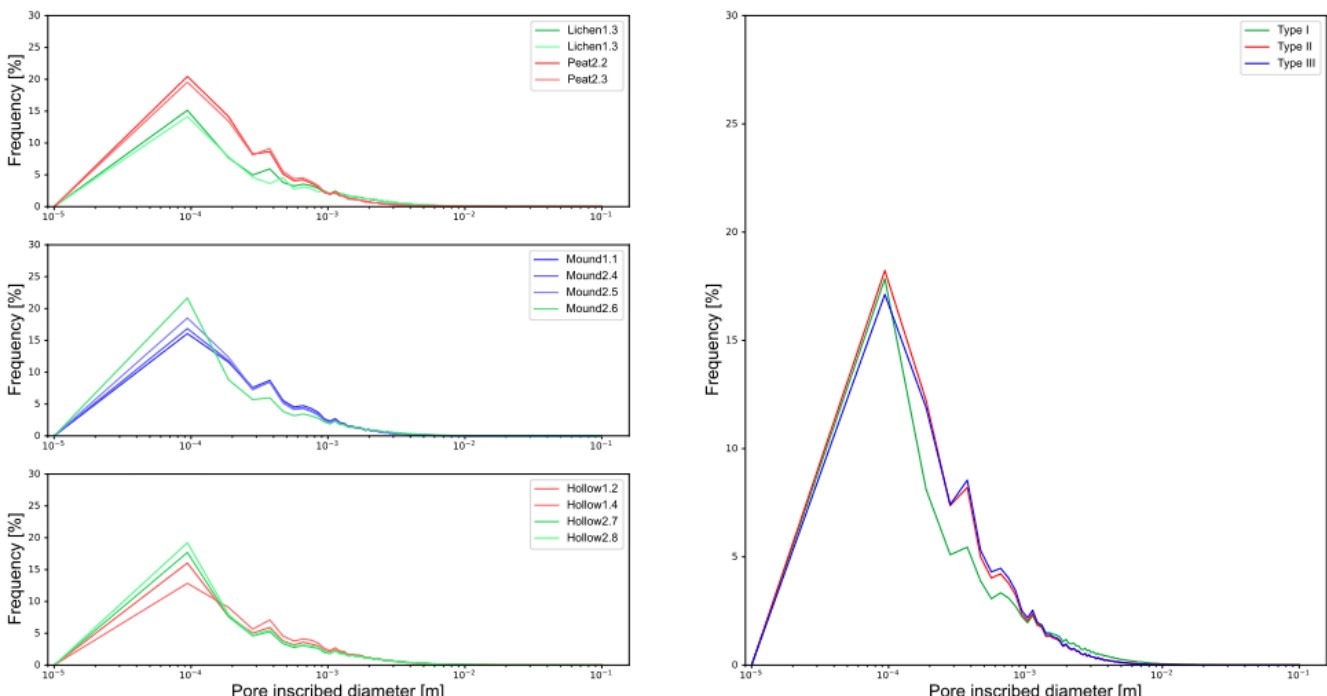

**Figure 5: Inscribed pore size distribution by classified type using particles' Feret diameter measurement. An averaged value is computed for each sample type, each color nuance representing each type.**

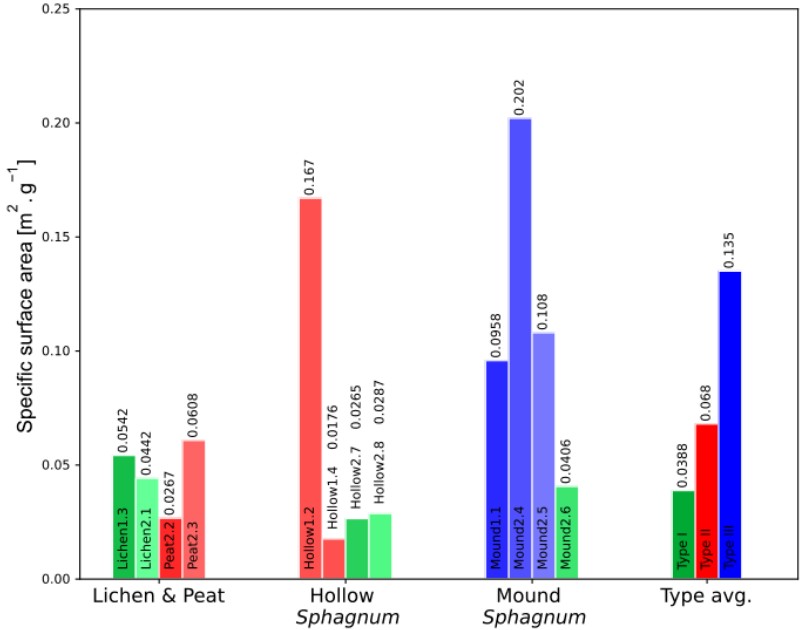

**Figure 6: Specific surface area [in m².g⁻¹] plots for each sample. Colors refer to each sample type.**



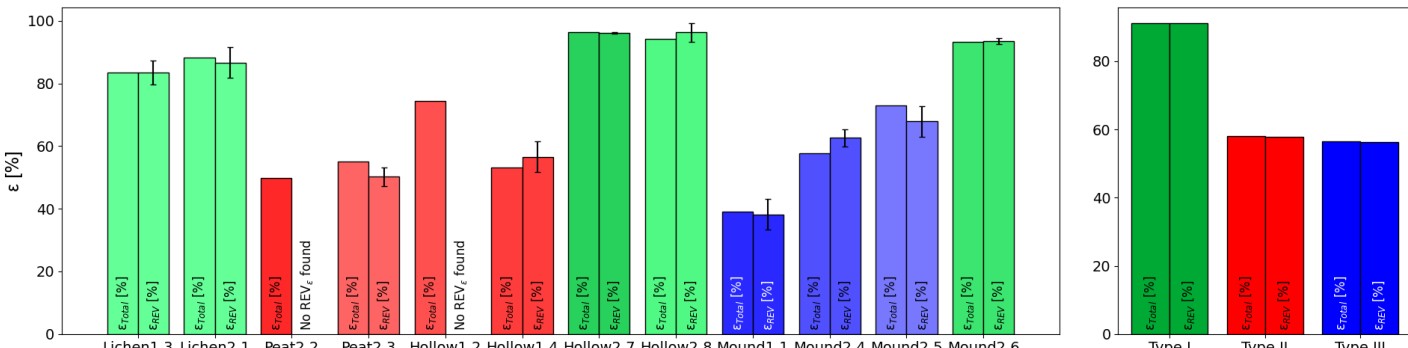

**Figure 7: Numerical porosity estimations [in %] for each sample. An averaged value is computed for each identified sample type (I, II, III) with corresponding color nuances. Peat 2.2 and Hollow1.2 did not admit any REV.**

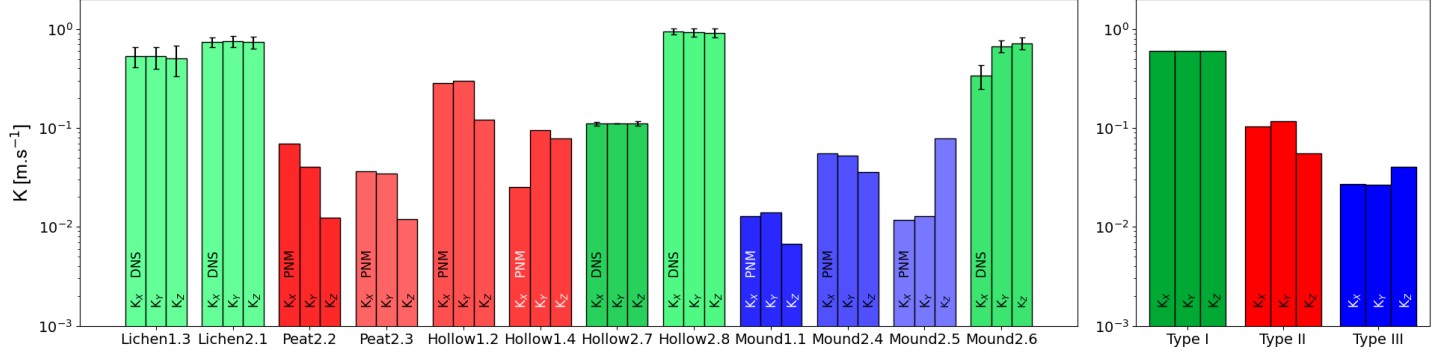

**Figure 8: Diagonal components of the hydraulic conductivity tensor (in m.s$^{-1}$) in $x$ ($K_{xx}$), $y$ ($K_{yy}$) and $z$ axis ($K_{zz}$) based on Direct Numerical Simulations (DNS) on Representative Elementary Volume of hydraulic conductivity (REV$_K$) for Type I samples and with a Pore Network Model (PNM) for Type II and III samples.**



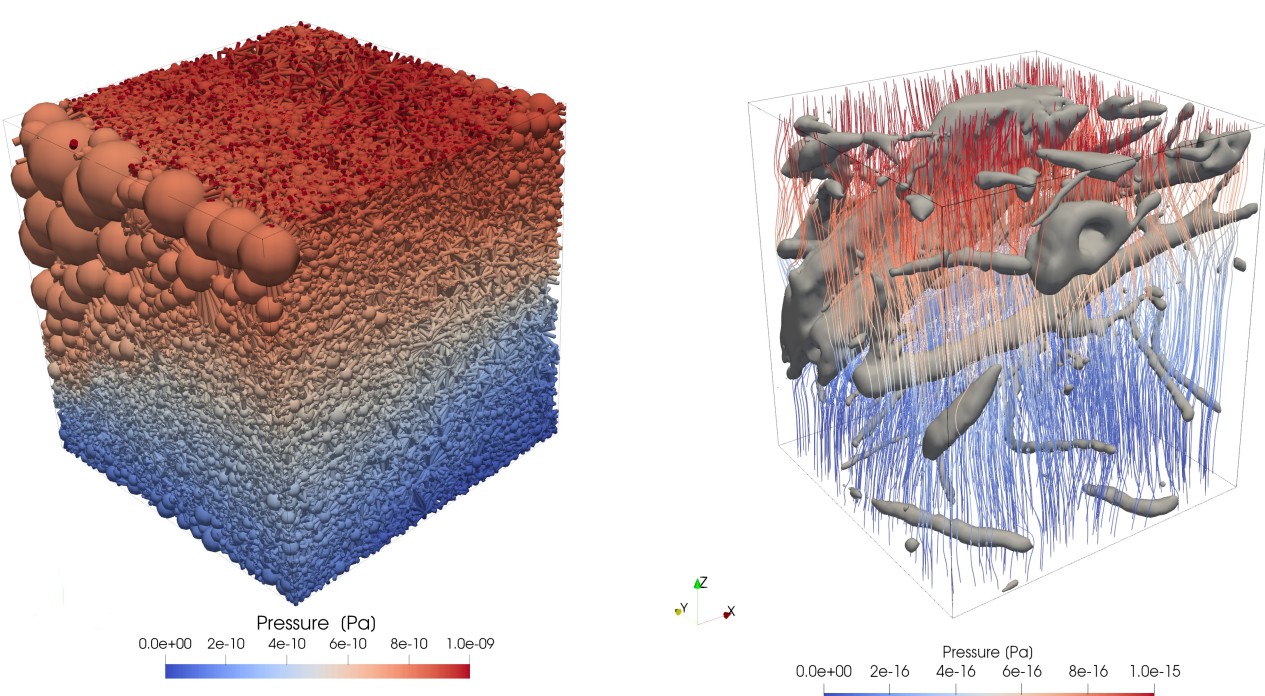

**Figure 9:** (*Left*) Pressure field (in Pa) after a single-phase flow simulation thorough a pore network model based on Mound2.5 sample. Spherical pore sizes are represented according to their respective size in the network. (*Right*) Pressure field lines (in Pa) after a single-phase flow simulation through a sub-sample of Hollow2.8 sample. The gray mesh corresponds to the isolated biological phase.



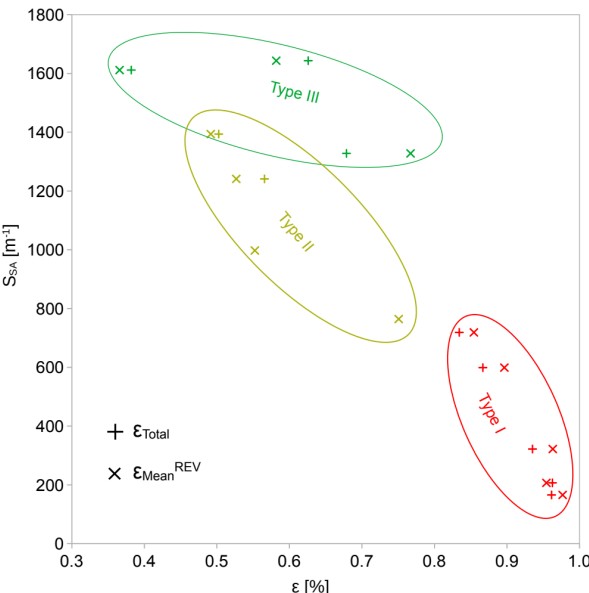

**Figure 10: Specific Surface (in m$^{-1}$) as a function of total porosity $\varepsilon_{Total}$ [%] and Representative Elementary averaged porosity $\varepsilon_{mean}^{REV}$ [%].**



**Table 1: Notation Glossary.**

| Symbol | Description | Unit |
|--------|-------------|------|
| Greek letters | | |
| $\rho_{dry}$ | Dried bulk sample density | $kg \cdot m^{-3}$ |
| $\rho_w$ | Water density | $kg \cdot m^{-3}$ |
| $\varepsilon_{total}$ | Sample digital overall porosity | % |
| $\varepsilon_{open}$ | Sample digital open porosity | % |
| $\mu_w$ | Dynamic viscosity of water | $Pa \cdot s^{-1}$ |
| $\sigma$ | Generic symbol for ratios | — |
| $\sigma_{S-T}$ | Ratio between sphere number and throat number (pore network) | — |
| Roman letters | | |
| $d_{Sph}$ | Spherical pore density (pore network) | $m^{-3}$ |
| $d_{Thr}$ | Throat density (pore network) | $m^{-3}$ |
| $g$ | Gravitational acceleration | $m \cdot s^{-2}$ |
| $i_{voxel}$ | Voxel intensity | — |
| $k$ | Intrinsic permeability | $m^2$ |
| $K_W$ | Hydraulic conductivity | $m \cdot s^{-1}$ |
| $L_i$ | Sample length along axis $i$ | m |
| $L_\varepsilon^{REV}$ | Representative Elementary Volume of porosity edge length | mm |
| $L_K^{REV}$ | Representative Elementary Volume of hydraulic conductivity edge length | mm |
| $N_b$ | Number of voxels of intensity N = b | — |
| $p_{open}$ | Ratio between $\varepsilon_{open}$ and $\varepsilon_{total}$ | — |
| $P$ | Pressure (water head) | Pa |
| Re | Reynolds number | — |
| $S_{outlet}$ | Overall surface of the outlet (including void and solid phase) | $m^2$ |
| $S_{SA(M)}$ | Mass specific surface area | $m^2 \cdot g^{-1}$ |
| $S_{SA(V)}$ | Volumetric specific surface area | $m^2 \cdot m^{-3}$ |
| $V_v$ | Volume of an array v | $m^3$ |
| $v_i$ | Incompressible flow speed along direction $i$ | $m \cdot s^{-1}$ |
| $u_i$ | Darcy flow along axis $i$ | $m^3 \cdot s^{-1}$ |





975 **Table 2: Synthesis of saturated hydraulic conductivity (in m.s⁻¹) of peat and *Sphagnum* found in the literature. *This study*'s values refer to field experiments conducted during sample collection (CHP: Constant Head Permeameter; FHP: Falling Head Permeameter; FP: Field Percolation; NM: Numerical Model (Hydrus-1D, see McCarter & Price, 2012)).**

| Sample | Method | $K_w$ [m.s⁻¹] | Reference |
|---|---|---|---|
| Peat | CHP | $4.6 \cdot 10^{-6} - 4.2 \cdot 10^{-4}$ | (Hamamoto et al., 2015) |
| Peat | CHP | $3 \cdot 10^{-5} - 1.3 \cdot 10^{-3}$ | (Quinton et al., 2000) |
| Sphagnum | CHP | $1.1 \cdot 10^{-2} - 4.3 \cdot 10^{-2}$ | (Golubev et al., 2021) |
| Sphagnum | Modified CHP | $2.4 \cdot 10^{-4} - 1.8 \cdot 10^{-3}$ | (Price et al., 2008) |
| Sphagnum | FHP | $1.2 \cdot 10^{-4} - 1.2 \cdot 10^{-3}$ | (Weber et al., 2017) |
| Peat | FP | $1.7 \cdot 10^{-6} - 3.3 \cdot 10^{-5}$ | (Päivänen, 1973) |
| Sphagnum | FP | $5.6 \cdot 10^{-5} - 1.7 \cdot 10^{-4}$ | (Crockett et al., 2016) |
| Sphagnum | NM | $2.9 \cdot 10^{-5} - 3.2 \cdot 10^{-3}$ | (McCarter & Price, 2012) |
| Sphagnum | FP | $6 \cdot 10^{-5} - 2 \cdot 10^{-4}$ | This study |

980 **Table 3: Computed global porosity ($\varepsilon_{total}$) and ratio of open porosity ($p_{open}$) for each sample obtained using a voxel counting algorithm.**

| Sample | $\varepsilon_{total}$ [%] | $p_{open}$ [%] | Class |
|---|---|---|---|
| Lichen1.3 | 83.5 | 0.9999 | Type I |
| Lichen2.1 | 88.2 | 0.9999 | Type I |
| Mound2.6 | 93.3 | 0.9999 | Type I |
| Hollow2.7 | 96.5 | 0.9999 | Type I |
| Hollow2.8 | 94.3 | 0.9999 | Type I |
| Hollow1.2 | 74.4 | 0.9984 | Type II |
| Hollow1.4 | 53.1 | 0.9980 | Type II |
| Peat2.2 | 49.8 | 0.9931 | Type II |
| Peat2.3 | 55.0 | 0.9938 | Type II |
| Mound1.1 | 39.1 | 0.9917 | Type III |
| Mound2.4 | 57.7 | 0.9900 | Type III |
| Mound2.5 | 72.9 | 0.9998 | Type III |



**Table 4: Obtained Representative Elementary Volume based on porosity (REV$_\varepsilon$). L$_\varepsilon^{REV}$ is the side length of a cubic REV of porosity. $\varepsilon_{mean}^{REV}$ is the average porosity of a given cubic REV. Ratio represents the volumetric percentage of the sample included in the REV.**

| Sample | L$_\varepsilon^{REV}$ [mm] | $\varepsilon_{mean}^{REV}$ [%] | Ratio [%] |
|---|---|---|---|
| Hollow1.2 | No REV | 74.4 | 100 |
| Hollow1.4 | 13.4 | 56.6 ($\pm$ 4.9) | 17.4 |
| Hollow2.7 | 2.82 | 96.1 ($\pm$ 0.3) | 3.3 |
| Hollow2.8 | 7.52 | 96.3 ($\pm$ 3.0) | 9.4 |
| Lichen1.3 | 1.88 | 83.4 ($\pm$ 3.9) | 2.1 |
| Lichen2.1 | 14.1 | 86.7 ($\pm$ 4.9) | 16.5 |
| Mound1.1 | 5.64 | 38.2 ($\pm$ 4.9) | 7.7 |
| Mound2.4 | 9.40 | 62.6 ($\pm$ 2.7) | 10.0 |
| Mound2.5 | 11.3 | 67.9 ($\pm$ 4.9) | 12.4 |
| Mound2.6 | 26.3 | 93.5 ($\pm$ 1.0) | 30.5 |
| Peat2.2 | No REV | 49.8 | 100 |
| Peat2.3 | 3.76 | 50.3 ($\pm$ 3.0) | 5.1 |

**Table 5: Diagonal components of the hydraulic conductivity tensor (in m.s$^{-1}$) for the studied Representative Elementary Volumes of hydraulic conductivity (REV$_K$) for type I samples using Direct Numerical Simulations.**

| Sample | L$_K^{REV}$ [mm] | K$_{xx}^{REV}$ [m.s$^{-1}$] | K$_{yy}^{REV}$ [m.s$^{-1}$] | K$_{zz}^{REV}$ [m.s$^{-1}$] |
|---|---|---|---|---|
| Hollow2.7 | 15.7 (167 vx) | $1.1 \cdot 10^{-1}$ ($\pm 5.0 \cdot 10^{-3}$) | $1.1 \cdot 10^{-1}$ ($\pm 6.07 \cdot 10^{-4}$) | $1.1 \cdot 10^{-1}$ ($\pm 5.1 \cdot 10^{-3}$) |
| Hollow2.8 | 15.7 (167 vx) | $9.5 \cdot 10^{-1}$ ($\pm 6.6 \cdot 10^{-2}$) | $9.3 \cdot 10^{-1}$ ($\pm 8.73 \cdot 10^{-2}$) | $9.1 \cdot 10^{-1}$ ($\pm 9.4 \cdot 10^{-2}$) |
| Lichen1.3 | 9.4 (100 vx) | $5.3 \cdot 10^{-1}$ ($\pm 1.3 \cdot 10^{-1}$) | $5.3 \cdot 10^{-1}$ ($\pm 1.31 \cdot 10^{-1}$) | $5.1 \cdot 10^{-1}$ ($\pm 1.7 \cdot 10^{-1}$) |
| Lichen2.1 | 15.7 (167 vx) | $7.4 \cdot 10^{-1}$ ($\pm 8.3 \cdot 10^{-2}$) | $7.5 \cdot 10^{-1}$ ($\pm 9.46 \cdot 10^{-2}$) | $7.4 \cdot 10^{-1}$ ($\pm 9.6 \cdot 10^{-2}$) |
| Mound2.6 | 11.8 (125 vx) | $6.8 \cdot 10^{-1}$ ($\pm 9.0 \cdot 10^{-2}$) | $6.7 \cdot 10^{-1}$ ($\pm 9.1 \cdot 10^{-2}$) | $7.2 \cdot 10^{-1}$ ($\pm 9.9 \cdot 10^{-2}$) |



**Supporting information**

**Supplement A: Global characteristics of collected samples' Usable Volume, numerical reconstructions and examples
of Representative Elementary Volumes of Porosity and Hydraulic conductivity. Species were identified according to
the morphological descriptions given in Volkova et al. (2018).** See file "SupplementA.pdf".

**Supplement B1: Overview of results of Representative Elementary Volumes of porosity for 10 of 12 samples (2 of
them did not converge to a solution). Convergence result for each sample is shown with a point and an error bar.** See
file "SupplementB1.pdf".

**Supplement B2: Overview of results of Representative Elementary Volumes of hydraulic conductivity for type I
samples. Each size matching the minimal standard deviation of diagonal hydraulic conductivity tensor is marked
with a "REV" sign.** See file "SupplementB2.pdf".

**Supplement C: Comparison between Direct Numerical Simulations (DNS) and Pore Network Modeling (PNM) for
Type I samples.** See file "SupplementC.pdf".