# Peer review of "Numerical Assessment of Morphological and Hydraulic Properties of Moss, Lichen and Peat from a Permafrost Peatland"

_EGUsphere, 2022_

## Author Response (AR1)

**Answering to Xiaoying Zhang**

Dear Xiaoying Zhang,

On behalf of all the co-authors, I wish to thank you for your careful review of this manuscript. Please find below the answers to the questions raised following the review. In order to make the answers accessible to everyone, we have decided to answer each question individually.

Sincerely yours,

Simon Cazaurang

General disclaimer: All the lines written in this answer are referring to the new revised version of the manuscript, which includes some supplemental information (highlighted in lime green) and some removed information (strike-through text in purple).

**General comment:**

[XZ]: This manuscript mainly presents a numerical assessment approach for the morphological and hydraulic properties of Western Siberian Lowland ground vegetation samples (Sphagnum moss, lichens, peat) by tomography scans. The numerical method based on digital X-CT recombination of samples can obtain the porosity and hydraulic conductivity. It provides a way to quantify hydrological properties of the bryophytic cover in permafrost-dominated peatland catchments. Overall, the contents of the manuscript are interesting. The logic of the paper is clear, and the results are well discussed and explained. However, there were some issues I was concerned about after I had read through the paper.

[Authors]: The authors want to thank the reviewer for his appreciation of our work. One can find below our detailed answers to the raised concerns. Each question is answered under the form of a paragraph regarding the raised question.

**Questions/Remarks:**

[XZ]: It seems quite complicated to obtain the porosity with the method the authors proposed compared to the traditional experimental method. In general, the advantage of using a non-destructive test is its coverage for large areas like remote sensing. However, with the method you proposed, you just measure the properties of samples as the traditional one and even more complicated. Thus, why does this method have superiority?

[Authors]: In this work, we wanted to prove that using a numerical scheme could provide us with data of the same stiffness and quality as traditional methods, which are here field percolation and constant-head permeameter. These methods have one major

disadvantage: they can only be performed a few times on a given sample. Using such methods is complicated when field study sites and quantification facilities are far from each other. Often, studies already available in the literature are only dealing with one property at a time for a given sample batch. Thus, experimental methods lead to a result, but they rely on a strong calibration in order to be statistically representative. Intercomparing experimental results is also intricate because experimental devices are different between studies.

Our samples are fragile, requiring extra care during sampling, transportation and analysis. Using non-destructive methods is interesting, even though it is true that our measurements heavily rely on samples' pre-processing after scanning, which is non-deterministic.

Beyond retrieving some values, we want to show through our study that numerical methods are elements of an innovative methodology that only relies on the input tomography. Indeed, numerical methods allow retrieving all usual properties at once using the sample batch, facilitating intercomparisons between them.

[XZ]: The authors stated that they confirmed the REV theory, but in the paper, I only see a schematic representation in Fig.2. Where is the data from the experiments?

[Authors]: A schematic representation of the REV theory is indeed available in Fig.2. Applying our method leads to computation of around 1.5 million of intermediate porosity values, with variable cube length. This process is done for each sample, creating 12 different graphs. Due to the numerous graphs and for the sake of clarity, we decided to put the data of the numerical experiments in Supplement C1 for porosity and in Supplement C2 for hydraulic conductivity. For each sample, the statistic spread as well as the mean porosity value is shown. The black point indicates the minimum REV identified, along with the property variability associated. We added a sentence to emphasize the presence of the results in Supplement C1 for porosity (line 292 in the revised manuscript) and in Supplement C2 for hydraulic conductivity (line 358 in the revised manuscript).

[XZ]: A schematic figure should be given to show the detailed methodology and technologies of *PNM*.

[Authors]: In our work, we used some specifically designed Pore Network Model libraries such as PoreSpy (https://porespy.org/) and OpenPNM (https://openpnm.org/) which are open-source and free. Both packages are relying on a dedicated algorithm named SNOW (Sub-Network of the Oversegmented Watershed). Some precision about the construction of such algorithm based on image processing is available in Gostick, 2017 (https://doi.org/10.1103/PhysRevE.96.023307). One can also refer to the following Python tutorial showing the construction of а network pore (https://porespy.org/examples/networks/tutorials/snow basic.html). Other pore network model generation algorithms are also existing in the literature (such as the "Maximal Ball" algorithm, Silin & Patzek, 2006). However, we decided to give a try to PoreSpy and OpenPNM because of their easy integration to the already existing workflow developed during the implementation of image processing. Many other pore network generators require extensive home-made codes, which makes it difficult to intercompare to other results from different materials.

[XZ]: The author stated the computation of K is represented in Fig.9, but what I see is a pressure field. What is the connection? And what is the relationship between a and b in Fig.9.

[Authors]: The authors warmly thank the reviewer for the careful reading of our manuscript. Unfortunately, this is a typo error in the figure call itself (line 480) . Indeed, Figure 9 shows pressure fields resulting from a single-phase flow simulation either through a Representative Elementary Volume of *Sphagnum* sample using Direct Numerical Simulation for the rightmost figure, or on a Pore Network for the leftmost figure. For both methods, we used a steady-state solver for Navier-Stokes equations. The pressure field was used to verify the flow equilibrium prior to permeability computations. Then, we used Darcy's law to compute intrinsic permeability, and then extensively hydraulic conductivity. In our work, we initially represented the velocity field instead of the pressure field. However, this choice was not optimal because velocity field visualization would require vector components. Our figures were then unreadable, as we wanted to show the sample's morphology.

[XZ]: As we all know, the porosity does not represent the condition of K. In the method, how did you distinguish the effective pores for the estimation of K? and decrease the impact or uncertainty that is caused by dead pores, as you are only counting the size of the pores?

[Authors]: In our method, we implement our simulation on pores' numerical representation, and not on a continuous medium. During morphological analysis, we showed that almost all pores are open and connected to each other. As the proportion of dead pores was often less than 0.1%, we made the assumption that all pores are effectively participating in flow dynamics, as we showed in Table 3.

[XZ]: For all the bar plots, I suggest you use different patterns instead of colors, as these colors are really hard to distinguish, especially in black and white print. Meanwhile, In Fig. 4 - Fig.8, one color represents different sample types, such as Lichen2.1, Hollow2.8, and Mound2.6. This can easily cause confusion. The same problem also appears in Fig. 5-Fig. 8. It is more appropriate to use one color to represent the same type of sample. Please modify the color of the samples again and add the description of types I- III in Fig. 4.

[Authors]: A new color and symbol scheme has been developed in the revised manuscript. Indeed, in the submitted version, we wanted to emphasize on the sample porosity types and not on the sample nature. Here, we added some symbols to cope with black and white prints.

[XZ]: A flowchart is better added to show how you estimate the n and K.

[Authors]: The authors would like to thank the reviewer for this suggestion. We added a flowchart (Fig. 3 in the revised version of the manuscript) describing the method to

estimate porosity Representative Elementary Volume. An analogous method is used for the quantification of the hydraulic conductivity K.

[XZ]: Line 110: "a thorough analysis of sample homogeneity is carried out, based on porosity...." and Line 245: "in the case of a homogeneous porous medium "and Line 25: "the most homogeneous samples" and "more heterogeneous samples", what is the criteria for judging whether the sample is "homogeneous"? Is there a mathematical relationship between the homogeneity of the sample and the porosity? And why do you have to emphasize the homogeneity. Does it mean the method is going to fail if the sample is "heterogeneous"? Is there a condition that the method is valid?

[Authors]: In our study, we assume that samples' variability is conditioned upon porosity spatially varying or not. This condition is a prerequisite, especially in the presence of natural samples, potentially very heterogeneous. Homogeneity criterion is based on the existence of a Representative Elementary Volume. Property homogeneity is required to find a Representative Elementary Volume. Consecutively, if a sample admits no significant variation for a given direction, then we can consider the sample homogeneous for that given axis (i.e., 1D-homogeneous). In the same way, if this low variability is observed on the three axes, then this sample is considered as homogeneous on the whole sample (i.e., 3D-homogeneous). Otherwise, convergence cannot be reached, and therefore the method using Representative Elementary Volumes will fail. To illustrate this, it is possible to take the example of samples Peat2.2 and Hollow1.2 that do not satisfy sufficient homogeneity to converge to a Representative Elementary Volume. In this case, apart from doing a computation on the entire sample (which is prohibitive in terms of computational costs), choices are reduced. For both samples, and more widely for the Type II and III samples that are 2D-homogeneous, Pore Network Modeling is more suitable in the absence of another efficient volume averaging method.

To enhance the classification, we added average and standard deviation values for each sample class in Table 3. This confirms that X and Y values have very low variability (under 1% for Type I samples, around 1% for Type III and less than 4% for Type II). In terms of vertical porosity, Type I have a low variability as Type III and Type II concentrates the variability.

[XZ]: In Fig. 2: the hydrological characteristics of lichen, Sphagnum moss, and peat are studied in this paper. However, Table 2 only collects references and hydraulic conductivity data for sphagnum and peat.

[Authors]: Indeed, there are only values for *Sphagnum* and peat in this table. As far as our literature review on lichen hydraulic properties went, we were only able to find one study about macroscale properties of lichens (Voortmann et al., 2014). The authors of this study found a value between 1.8.10-9 m.s-1 to 3.7.10-9 m.s-1 using an evaporation experiment and inverse modelling. We added this value to Table 2 among the other values for *Sphagnum* moss and peat. However, we express reservations about the representativeness of the obtained value. Indeed, the morphology of the presented samples seems close to those we

are studying. The hydraulic conductivity remains nevertheless weak with regard to the observed structure. It would be interesting to apply the numerical scheme developed in the present manuscript in order to compare the results.

[XZ]: Line 403-405: Specific surface area of the sample obtained using the PNM method is always larger than those obtained by image processing. Besides the explanation from the perspective of porosity, are there any other reasons to prove this phenomenon? Is it true? What is the reason? And how does the specific surface area affect the estimation of K?

[Authors]: Even though Pore Network Models are computationally more efficient than Direct Numerical Simulations, they rely on a unidimensional reconstruction of a tridimensional structure. This reconstruction causes structural simplifications and potentially errors.

Specific surface area is important for chemical and thermal reaction studies. They require in essence the most high-fidelity surface representation. Finding a specific surface area from a simplified model does not have much sense for comparison. Indeed, the authors have chosen to remove the comparison between specific surface obtained by image processing and pore network models (line 447 of the manuscript).

**[XZ]: Is the application promising like what I proposed in the beginning? Why is this hard work worthy if your only goal is to get the same values ?**

[Authors]: In our point of view, even though the implementation of such numerical schemes are indeed complex and time-consuming, we believe that using numerical methods enables a better comparison between samples of different locations. Morphological and hydraulic properties assessments based on field or laboratory experiments have each one their own biases. For these methods, we can identify some possible biases :

- 1. Date and time of the sampling;
- 2. Experimenter and his/her skills or knowledge;
- 3. Transport conditions;
- 4. Physical and biological degradation;
- 5. Used laboratory hardware to achieve these measurements;
- 6. Data post-processing.

Field experiments also have some limitations. For example, a double-ring infiltrometer test was conducted during the sampling field trip next to each sample plot. In this case, the experiment was impossible to conduct for some samples. This links well with our results that show high hydraulic conductivity.

When using a numerical scheme, computations can be virtually done as many times as needed. Each computation attempt will produce the same result for a given input dataset. We can also eliminate biases #4 and #5 as the tomography gives us a steady-shot of a given

sample, allowing us to study these samples thoroughly, only relying on the minimal resolution of the X-ray scanner of 88  $\mu$ m.voxel-1.

We found that using numerical schemes enables classifying samples along their homogeneity or heterogeneity, which is not easy if based on experiments only. Our method is valid even for heterogeneous samples on the condition that sampling and sampling have been conducted thoroughly. A small paragraph has been added in the conclusion (from line 601 down to line 608) to emphasize these points.

Our method allowed to pinpoint the importance of microscale characterization in order to understand macroscale phenomena. We showed that applying an analogous method is potentially valid for all kinds of porous media, even heterogeneous. Such kind of study has not been done so far to our knowledge.

Finally, we assume that the implemented numerical scheme is efficient compared to field and laboratory experiments because such work would require a statistically strong amount of samples. We compensate the low number of samples by the REV survey that validates sufficient statistical representativity of our samples. Collecting such numerous samples is difficult, both for transportation and preservation of this sensitive interface.

**Minor remarks:**

[XZ]: In Table D1, please add the specific surface area and porosity data of samples obtained from image processing.

[Authors]: We acknowledge this remark. We included these values in the revised version of our manuscript.

[XZ]: In Table D4, Please check whether the symbols are correct, such as dSph, σS-T, and dThr. [Authors]: We acknowledge this remark. The labels of Table D1 have been modified accordingly.

[XZ]: In the abstract, summary and conclusions, the authors should add supplement contents about the limitations of this proposed model method, as well as the scientific importance of this study.

[Authors]: The authors thank the reviewer for this suggestion. Indeed, we gave some details about the limitations of the proposed model method. Indeed, the main limitation observed in our method is the input data (which is here the tomographic scans). This study highlights the need of a quantitative and reproducible microscale properties' assessment in order to implement values in a macroscale model. Our combined method enable to study any type of sample, even heterogeneous ones. Such inclusive method has not been seen elsewhere by the authors. We modified the essence of the abstract with a dedicated paragraph from line 39 to 44). We also added some information in the conclusion (from line 601 down to 609).

**Answering to Anonymous Reviewer**

Dear Madam, dear Sir,

On behalf of all the co-authors, I wish to thank you for your careful review of this manuscript. Please find below the answers to the questions raised following the review. In order to make the answers accessible to everyone, we have decided to answer each question individually.

Sincerely yours,

Simon Cazaurang

General disclaimer: All the lines written in this answer are referring to the new revised version of the manuscript, which includes some supplemental information (highlighted in lime green) and some removed information (strike-through text in purple).

**General comment:**

The submitted manuscript deals with the assessment of the porosity and hydraulic conductivity of Western Siberian Lowland ground vegetation samples (lichens, Sphagnum mosses, peat). Twelve samples are analysed throughout a numerical method instead of a classical experimental field determination (that they have also carried out). Based on digital X-CT reconstructions the study confirms the high values of porosity presented by such a biological media. Due to numerical constraints, a single numerical method could not be implemented and the authors used both Direct Numerical Simulations (DNS) and Pore Network Modeling (PNM), which did not provide the same results... but are closer together for the evaluation of the hydraulic conductivity, which is several orders of magnitude lower for the experimental field method.

The subject of the study is interesting and suitable for publication in HESS and as the authors reported in their introduction, the results could be of great importance to model the evolution of these natural environments impacted by climate change. The manuscript is globally well written and of good scientific quality. I think that a few modifications can still be made to improve its quality.

- *References are numerous, and I guess that they cover the state of the art.*
- The purpose of the study should be more clearly stated.
- The station can be described briefly, the field work is important and deserves a few words.
- A complete flowchart describing the different steps of the methodology will be appreciated.
- The quality of the figures is good, but the use of colors is not always clear and understandable. The works in the supplements are not highlighted enough.

- The distinction between homogeneous and heterogeneous is not clearly defined, although this aspect seems to have an impact on the typology and the possibilities of implementing the proposed methods.
- Some misunderstandings in the presentation of the results for porosity.
- The definition of the sample typology is done twice and is not consistent (?)

[Authors]: The authors want to warmly thank the reviewer for the careful and thorough reading of this manuscript. One can find the answers to the raised comments below. Each question is answered through a dedicated paragraph for the question.

**Comments:**

[AR] L 97: I am not a specialist of the topic, but what about the study by Potkay et al. (2020)?

Potkay, A., ten Veldhuis, M. C., Fan, Y., Mattos, C. R., Ananyev, G., & Dismukes, G. C. (2020). Water and vapor transport in algal-fungal lichen: Modeling constrained by laboratory experiments, an application for Flavoparmelia caperata. Plant, cell & environment, 43(4), 945-964.

[Authors]: The authors want to thank the reviewer for this suggestion. There are some publications about lichen membrane transport properties assessment and modelling (Linefold et al., 1990, Voortmann et al. 2014 and for instance Potkay et al., 2020). However, most of these publications are dealing with transmembrane transport and not macro-scale transport. Nonetheless, we added Potkay et al. (2020) for the sake of completeness of the state of the art (line 108). Making the link between transmembrane transport and macro-scale transport is a significant clue for understanding the whole atmosphere-geosphere transfer phenomenon.

[AR] L105: what is the main purpose of the present study? The authors want to assess hydraulic properties of lichens and Sphagnum mosses by numerical methods, but I do not understand the "justification" of this way? What kind of problems / difficulties arise from the experimental measurements? The question is of primary goal and I think that the issue of measurements scale should also be mentioned, especially if the results are intended to be used in a modelling approach.

[Authors]: Indeed, we do not emphasize enough the main purpose of the study. The authors want to thank the review to penlight this aspect. We added a supplementary paragraph (lines 110 to 124) to make the link between the state of the art and the main workflow applied for this study, insisting on the gains in terms of reproducibility and inter-comparison capability.

[AR] L122: please indicate the coordinates of the station where samples were collected. Besides, I would appreciate synthetic details on the climatic condition of this location.

[Authors]: The authors added some supplementary information about the scientific station where the samples were collected and more broadly on northern taiga (lines 146-148).

Some details are available through various publications such as Payandi-Rolland et al. (2020), Raudina et al. (2018), Soudzilovskaïa et al. (2013).

[AR] L161: you refer to the drying experiment carried out by Kämäräinen et al. (2018). A different drying temperature (40 °C instead of 50 °C) has been applied for the preparation of samples that have a larger size in your case. It is not exactly the same protocol. The most important finding is that the morphological structure is preserved, and since you don't expect to know the actual water content of your various samples, this is probably not a critical point.

[Authors]: Actually, the sentence is not well-written. It would be more accurate to write "an analogous method than the one used in Kämäräinen et al. (2018)" rather than saying that it is the same method. We corrected this sentence (line 188) in the revised version of the manuscript.

[AR] L190: in the main text, Fig. 4 is used before figures 2 and 3? It is quite difficult to distinguish colors in your picture. Is there a link between the colors used for your planar porosity plots and the averaged versions? I don't think so, but it's not clear.

[Authors]: We intended to highlight samples' homogeneity classes through the proposed color scheme. However, this layout was not optimal for black and white prints. A new color palette has been developed to cope with black and white prints and to enhance readability for samples' nature and homogeneity classes. A sentence has been added in the revised manuscript to emphasize on the manner the color scheme was though out (line 223).

[AR] L202: please indicate that  $p_{open}$  refers to the open porosity proportion. In table 3, this variable should be multiplied by 100? And to be consistent with equation (2), you should indicate "%" in Table 1.

[Authors]: The authors want to thank the reviewer for the rigorous reading. We made the choice of keeping  $p_{open}$  as a decimal number to avoid confusion with other percentages (especially porosity). We corrected equation (2) accordingly to what is shown in Table 1 and the results of Table 3.

[AR] L258: You should refer to Supplement A. Also, I think you did not mention in the main text supplement B1 where some pictures present the evolution of porosity with the size of measurement. I'm wondering how you fix the final value of the sample's REV of porosity. It is not obvious to understand your results (final sizes) when comparing curves, for instance, for samples Hollow2.7, lichen1.3 and lichen2.1.

[Authors]: Indeed, there are missing links to Supplement A and C1. We made the links between the Supplements and the main text clearer for the reader.

Assessing the REV maximal size is one of the main challenges in REV computations. We presented REV in part 2.4 of the manuscript. Yet, we reiterated the criteria in this subsection as well as in Supplement C1's caption.

The final REV of porosity size is computed according to the standard deviation values for each study size reduction. Each time when the standard deviation overpasses a certain arbitrary threshold (here, we have decided to take 1, 3 or 5% of porosity variation). In the

case of Hollow2.7, Lichen1.3 and Lichen2.1, we observed that these samples are more homogeneous than the others.

To enhance the classification, we added average and standard deviation values for each sample class in Table 3. This confirms that X and Y values have very low variability (under 1% for Type I samples, around 1% for Type III and less than 4% for Type II). In terms of vertical porosity, Type I have a lower variability than Type II and Type III, the latter types concentrating porosity variability.

**[AR] L265: I don't think that figure 3 is required.**

[Authors]: Indeed, permeability measurements using pressure gradients (constant and variable head permeameters) are often used in the literature. However, we assume that no standardization is made for such numerical resolution. In this way, we wanted to emphasize boundary and initial conditions to clarify numerical aspects. We moved this figure to the supplemental section, as it is not mandatory for the manuscript itself.

[AR] L280: I guess the numerical method developed by Patankar (1980) has been improved to solve faster the single-phase flow problem. I understand that the numerical aspects are not a key point of this study, but your choice induces limitations in the sample processing capacity (only type I) and that's a bit bothering.

[Authors]: The actual solver to conduct our computations is *simpleFoam*, built on the open-source computational fluid mechanics toolbox OpenFoam. The *simpleFoam* solver is far apart from Patankar's original algorithm. Nonetheless, *simpleFoam* still uses the same numerical solving scheme to resolve Navier-Stokes equations, therefore we thought that it was important to cite Patankar (1980). *simpleFoam* is nowadays highly parallelizable on many processors so that even big simulations can be conducted fairly easily. In our case, the number of simulations per sample (672) is rather the most limiting factor than the cost of the simulation itself.

However, the sentence could lead to misunderstand our methodology. We corrected this sentence in the revised version of the manuscript (line 317).

[AR] L282: how did you select your 4 REV sizes? If I clearly understand, choosing a small REV size involves performing many simulations. However, are 4 sizes enough to detect the impact of the size on the fluctuation of your variable?

[Authors]: In this study, we assessed hydraulic conductivity REV using an analogous method of the one developed for the REV of porosity. Nonetheless, for the hydraulic conductivity REV, the sizes are assigned from the beginning to shorten computational times. Indeed, this study took 10 days per sample to be conducted. Making a continuous scanning as made for porosity is still prohibitive in terms of computational efficiency.

The sentence on line 282 can be misleading. We reformulated this sentence in the revised version (line 323).

[AR] L344-354: The conclusion is that DNS and PNM did not lead to compatible results for hydraulic conductivity measurements. The authors should probably indicate a way to - a priori - select the better method for each sample. Is there a link with homogeneity of the sample (the authors have indicated that in the manuscript, L276 for instance) and how can they mathematically or physically define the frontier between homogenous and heterogeneous sample?

[Authors]: Indeed, our study spotlights the fact that it is not possible to study a sample in the same way if there is homogeneity or heterogeneity. In our study, we assume that samples' variability is conditioned upon porosity spatially varying or not. This condition is a prerequisite, especially in the presence of natural samples, potentially very heterogeneous. Homogeneity criterion is based on the existence of a Representative Elementary Volume. Property homogeneity is required to find a Representative Elementary Volume. Consecutively, if a sample admits no significant variation for a given direction, then we can consider the sample homogeneous for that given axis (i.e., 1D-homogeneous). In the same way, if this low variability is observed on the three axes, then this sample is considered as homogeneous on the whole sample (i.e., 3D-homogeneous). Otherwise, convergence cannot be reached, and therefore the method using Representative Elementary Volumes will fail. To illustrate this, it is possible to take the example of samples Peat2.2 and Hollow1.2 that do not satisfy sufficient homogeneity to converge to a Representative Elementary Volume. In this case, apart from doing a computation on the entire sample (which is prohibitive in terms of computational costs), choices are reduced. For both samples, and more widely for the Type II and III samples that are 2D-homogeneous, Pore Network Modeling is more appropriate in the absence of another efficient volume averaging method.

To enhance the classification, we added average and standard deviation values for each sample class in Table 3. This confirms that X and Y values have very low variability (under 1% for Type I samples, around 1% for Type III and less than 4% for Type II). In terms of vertical porosity, Type I have a low variability as Type III and Type II concentrates the variability.

[AR] L349: Fig. 9-Right does not describe the hydraulic conductivity computation as mentioned in the manuscript.

[Authors]: Indeed, this is a typo error. We corrected this sentence in the manuscript (line 495).

[AR] :

- *L359: the lowest porosity value is obtained for the Mound1.1 sample.*
- L365: according to Table 3, the average porosity for mound mosses is 65.8 ± 23%?
- L373: Lichen1.3 has a total porosity lower than 85%.
- L375: correct definition is "medium high porosity" or "low basal porosity" (L193) for type II ? Besides, total porosity is not comprised between 70 and 85%.
- L376: for type III, the total porosity is lower than 73%

[Authors]: The authors want to thank the reviewer for the careful reading and verification of the data. We corrected promptly these remarks in the revised version of the manuscript.

For mound mosses, the average porosity is  $65.8 \pm 23\%$ , including Mound2.6. As the value span is quite significant, we reworked the type definitions to be consistent with the definitions given on lines 219-222.

[AR] L469: Did Shirokova et al. (2021) make a link between their biological results and the properties you focused on? I don't think that they talk about porosity, hydraulic conductivity, and specific surfaces. Either you add a reference that show this link, or you add this part later in your discussion...

[Authors]: The citation of Shirokova et al. (2021) is not well inserted in our discussion, even though this paper is important because our current study and their study are well linked. Indeed, transmembrane transport properties are required to make a realistic flux and matter balance through arctic vegetation cover. For the sake of clarity, this citation has been moved in the introduction and the linkage has been reinforced.

[AR] L472: your experimental results are quite similar than other studies mention in Table 2. I would include them (not only "published results" but experimental results in general, including your own).

[Authors]: The authors want to thank the reviewer for this suggestion. We added our experimental results for each sample in the manuscript. We decided to add the results of the field work in the discussion part (Table 6) to give some material to discuss the differences between field and numerical experiments. The authors want to thank Georgiy Istigechev for his help in the double-ring infiltrometry campaign. We added him in the contributions as well as in the acknowledgements.

[AR] L477: What is the interest of subsection "4.1 Numerical reconstruction after scanning" in the discussion part? I have the feeling that the main elements are given again between L 510 and 520?

[Authors]: Section 4.1 describes the drawbacks that were identified from the use of the tomographic plots themselves. Then, between lines 510 and 520, we introduce the differences between DNS and PNM. In our point of view, we think that it is more appropriate to keep these aspects separated, as they are occurring at a different moment in the process.

[AR] L523-526: I think the main justification and interest of the present study lies in the fact that field experiments carried out to obtain hydraulic conductivity could be inaccurate because of an excessive compression of the biological media. I would appreciate references that could highlight this aspect, also the difference between natural rainfall and field experiments and finally the lack of such kind of experimental measurements. Besides, in the perspective of numerical modelling of these biological media located in the upper part of the soil, is it possible to hide unsaturated flows (cf. L488)? I guess it's not possible to directly measure water flow, but do you know of any experiments or field measurements where high velocity is consistent with your very high hydraulic conductivity?

[Authors]: Some few studies deal with compressibility issues occurring in the field measurement (for instance, Weber et al., 2017 and Golubev & Whittington, 2018). These references have been added accordingly in the text. Moreover, this study is a preliminary step of a broader study of transfer properties of this biological media. Here, we assessed fully saturated hydraulic conductivity, which is something occurring in extreme conditions. Hydraulic conductivity will be studied for variably saturated porous media and will be the subject of another study out of the scope of this study.

Minor comments:

L52 "increases"  $\rightarrow$  **Done**

**OK** L93, L109 and in Table 2:Ref. Hamamoto et al. à Date of publication is 2016  $\rightarrow$  **Done**

OK L159: IMFT  $? \rightarrow$ **Done**

OK L386: 1.00 mm  $\rightarrow$  **Done**

OK L722: "response" a typo in the original title?!  $\rightarrow$  **Done**

OK L591: problem with the words in italics "Sphagnum"  $\rightarrow$  **Done**

OK L690: problem with the words in italics "Sphagnum"  $\rightarrow$  **Done**

OK L770:Date of publication is 2012 instead of 2014 ? (it is correctly mention in the main text: "McCarter & Price, 2012")  $\rightarrow$  Done

OK L840-841: problem with subscript and italics in the reference title "Extending a land-surface model with Sphagnum moss to simulate responses of a northern temperate bog to whole ecosystem warming and elevated  $CO2'' \rightarrow Done$